# Graph Generative Pre-trained Transformer

**Xiaohui Chen** [1]  **Yinkai Wang** [1]  **Jiaxing He** [2]  **Yuanqi Du** [3]  **Soha Hassoun** [1]  **Xiaolin Xu** [2]  **Li-Ping Liu** [1]

## Abstract

Graph generative models, which can produce complex structures that resemble real-world data, serve as essential tools in domains such as molecular design and network analysis. While many existing generative models rely on adjacency matrices, this work introduces a token-based approach that represents graphs as token sequences and generates them via next-token prediction, offering a more efficient encoding. Based on this methodology, we propose the Graph Generative Pre-trained Transformer (G2PT), an autoregressive Transformer architecture that learns graph structures through this sequence-based paradigm. To extend G2PT's capabilities as a general-purpose *foundation* model, we further develop fine-tuning strategies for two downstream tasks: goal-oriented generation and graph property prediction. Comprehensive experiments on multiple datasets demonstrate G2PT's superior performance in both generic graph generation and molecular generation. Additionally, the good experiment results also show that G2PT can be effectively applied to goal-oriented molecular design and graph representation learning. The code of G2PT is released at https://github.com/tufts-ml/G2PT.

## 1. Introduction

Graph generation has emerged as a crucial task across diverse fields such as chemical discovery and social network analysis, thanks to its ability to model complex relationships and produce realistic, structured data (Du et al., 2021; Zhu et al., 2022).

Early generation methods such as DeepGMG (Li et al., 2018) and GraphRNN (You et al., 2018b) model graphs with sequential models. These approaches employed sequential frameworks (e.g., RNNs or LSTMs (Sherstinsky, 2020)) to generate graphs sequentially. For instance, GraphRNN generates adjacency matrix entries step by step. For undirected graphs, it only needs to generate the lower triangular part of the adjacency matrix. DeepGMG frames graph generation as a sequence of actions (e.g., add-node, add-edge), and utilizes an agent-based model to learn the action trajectories.

Recent advances in graph generative models have primarily focused on permutation-invariant methods, particularly diffusion-based approaches (Ho et al., 2020; Austin et al., 2021). For example, models like EDP-GNN (Niu et al., 2020) and GDSS (Jo et al., 2022a) learn from adjacency matrices as continuous values. DiGress (Vignac et al., 2022) and EDGE (Chen et al., 2023) employ discrete diffusion, treating node types and all node pairs (edges and non-edges) as categorical variables. These models start from a random or a fixed adjacency matrix and run "denoising" steps to sample an adjacency matrix from the target graph distribution. They specify exchangeable (permutation-invariant) distributions over graphs by assigning the same probability to adjacency matrices of the same graph. However, achieving the permutation-invariant property has a price: the underlying neural network needs to be permutation-invariant as well, limiting the architecture choice to graph neural networks only. Discrete diffusion has an additional limitation: it samples matrix entries independently at each denoising step, making it challenging to learn the true distribution when the number of denoising steps is insufficient (Lezama et al., 2022; Campbell et al., 2022).

In recent years, the revolutionary success of large language models (Achiam et al., 2023; Dubey et al., 2024) shows the power of autoregressive Transformers and also inspired the application of these models in other fields such as image generation (Esser et al., 2021). In this work, we revisit the sequential approach to graph generation and introduce a novel token-based encoding scheme for representing sparse graphs as sequences. This new encoding strategy unlocks the potential of Transformer architectures for graph generation. We train autoregressive Transformers to generate graphs by predicting token sequences, resulting in our proposed method: Graph Generative Pre-trained Transformer (G2PT). While G2PT does not maintain permutation invariance, we argue that its capacity to learn accurate graph

---

[1]Tufts University [2]Northeastern University [3]Cornell University. Correspondence to: Xiaohui Chen <xiaohui.chen@tufts.edu>, Li-Ping Liu <liping.liu@tufts.edu>.

*Proceedings of the $42^{nd}$ International Conference on Machine Learning*, Vancouver, Canada. PMLR 267, 2025. Copyright 2025 by the author(s).

| Model (Rep.) | Likelihood | Illustration | #Network Calls | #Variables | $p(\mathbf{A})$ or $p(E)$ |
|---|---|---|---|---|---|
| Diffusion($\mathbf{A}$) | $p(\mathbf{A}^T)\prod\limits_{t=1}^{T}p(\mathbf{A}^{t-1}\|\mathbf{A}^t)$ |  | $T$ | $O(Tn^2)$ | Intractable |
| Sequential($\mathbf{A}$) | $\prod\limits_{i=2}^{n}\prod\limits_{j=1}^{i-1}p(\mathbf{A}_{i,j}\|\mathbf{A}_{<i,<i-1},\mathbf{A}_{i,<j})$ |  | $O(n^2)$ | $O(n^2)$ | Full factorization |
| Sequential($E$) | $p(e_1)\prod\limits_{i=2}^{m}p(e_i\|e_{<i})$ |  | $O(m)$ | $O(m)$ | Full factorization |

*Table 1.* Families of graph generative models and their graph representations. Here $n$ is the number of nodes, and $m$ is the number of edges. The illustration use solid and dash lines to represent edges and non-edges respectively. The (non-)edges generated at current step are in blue. Our proposed G2PT is an autoregressive model and learns with Sequential($E$).

distributions from large-scale data outweighs this limitation. Table 1 gives a comparison of diffusion-based models, sequential models based on adjacency matrices, and our model based on edges.

This new approach offers an additional advantage: it allows us to leverage recent training techniques developed for Transformers in NLP. Specifically, we apply our model to goal-oriented generation tasks, such as optimizing molecular properties. To this end, we explore both rejection sampling fine-tuning and reinforcement learning, each designed to increase the probability mass assigned to desirable graphs. Additionally, our model can be used to learn graph representations when provided with the entire graph as input – a capability not feasible with diffusion-based generative models. With fine-tuning and a supervised objective, G2PT can also be adapted for graph-level tasks such as graph classification.

We evaluate G2PT on a series of graph generation tasks: generic graph generation, molecule generation, and goal-oriented molecular generation. Without excessive architectural engineering or training tricks, G2PT performs better than or on par with previous state-of-the-art (SOTA) baselines over seven datasets. By fine-tuning G2PT towards generating molecules with target properties, we showcase that G2PT can be easily adapted to various generative tasks that require additional alignment. We also fine-tune G2PT for molecular property prediction on MoleculeNet datasets. The results demonstrate the effectiveness of G2PT's learned representations for classification tasks. Finally, we analyze the G2PT model and show its performance with data augmentation.

**Contributions.** Our main contributions are as follows:

- We propose a novel token-based graph representation that enables efficient graph generation;
- We introduce G2PT, a Transformer decoder trained on the new graph representation to model sequence

distributions via next-token prediction;
- We explore fine-tuning techniques to adapt G2PT for downstream tasks, such as goal-oriented graph generation and graph property prediction;
- We conduct an extensive empirical study of G2PT, which achieves strong performance across diverse graph generation and prediction tasks.

## 2. Related work

**Permutation-invariant vs sequential models.** Neural graph generative models are first developed as autoregressive models. These models represent graphs as sequences and learn to generate such representations. A variety of models such as GraphRNN (You et al., 2018a), GRAN (Liao et al., 2019), GraphDF (Luo et al., 2021), and DAGG (Han et al., 2023) generate entries of the lower triangular of the adjacency matrix of the target graph, while DeepGMG (Li et al., 2018) and BiGG (Dai et al., 2020) directly generate edges through node representations learned from partially generated graphs.

In parallel, another research direction considers permutation-invariant models to predict logits of adjacency matrices and then sample the adjacency matrix in one step (Madhawa et al., 2019; Liu et al., 2019). These models define exchangeable distributions over graphs. This approach has been further advanced with diffusion-based methods (Ho et al., 2020; Austin et al., 2021), which can be divided into continuous diffusion models (Niu et al., 2020; Jo et al., 2022b) and discrete diffusion models (Vignac et al., 2022; Chen et al., 2023; Qin et al., 2024). Among these, discrete diffusion models have shown superior performance due to their alignment with the discrete nature of graphs. These models begin with a fixed or random adjacency matrix and iteratively update matrix entries over multiple steps to converge to the target adjacency matrix. Each update step, referred to as a "denoising" step, modifies a subset of the adjacency matrix entries.

Discrete diffusion graph models typically require many denoising steps for graph sampling. This is because entries are sampled independently conditioned on the previous step for each step. Such a sampling scheme introduces the compounding decoding error (Lezama et al., 2022), leading to a poor approximation to the true distribution (Campbell et al., 2022).

**Generating adjacency matrices vs edge lists**. Most graph generative models, including the majority of sequential approaches (You et al., 2018b; Liao et al., 2019) and all diffusion-based methods (Jo et al., 2022b; Niu et al., 2020; Vignac et al., 2022; Chen et al., 2023; Qin et al., 2024), generate graphs via their adjacency matrix, with a few exceptions (Li et al., 2018; Dai et al., 2020) that operate directly on the edge list of the target graph.

Despite the popularity, learning adjacency matrices requires modeling all node pairs in a graph, and thus needs a computation quadratic in the number of nodes $n$. For autoregressive models, it creates long sequences for the model to learn, making it a challenging learning problem (Hihi & Bengio, 1995). Therefore, generating adjacency matrices are computationally intensive for both autoregressive and diffusion-based models. In contrast, edge lists only define variables over actual edges. Compared to adjacency matrices, models that use edge lists usually require less computation for both learning and sampling, especially when dealing with sparse graphs.

Despite the advantage mentioned above, modeling edge lists has received limited attention because of the difficulty of modeling structurally heterogeneous discrete events (e.g., adding a node and deciding the two ends of an edge). Li et al. (2018) propose an action-based framework and model a graph with pre-defined actions such as "add-node" and "add-edge". With a special design of the model architecture, the model is challenging to train. Additionally, the "add-edge" action cannot be treated as label prediction because the number of nodes increases with the generation procedure. Previous methods rely on node representations learned from graph neural networks, which are shown to have limited expressiveness in general (Li & Leskovec, 2022). This work takes a different approach and designs a token-based sequences that enable generic Transformer architectures.

## 3. Graph Generative Pre-trained Transformer

### 3.1. Representing Graph as Sequence

We consider modeling a graph as a sequence that first lists all nodes and then all edges. Let $G = (V, E)$ denote a graph where nodes and edges are each associated with a type label. Here $v \in V$ is represented as a tuple $v := (v^c, v^{id})$, where $v^{id} \in \mathbb{Z}^+$ is the node index and $v^c \in \{1, \ldots, K_v\}$ is the node type. And $e \in E$ is represented as a triple $e :=$

---

**Algorithm 1** Degree-Based Edge Removal Process
***
**Input:** Graph $G = (V, E)$, neighborhood function $\text{Nei}(\cdot)$
**Output:** Sequence of removed edges $\sigma_E$
Initialize $\sigma_E \leftarrow [\,]$
**while** $E \neq \emptyset$ **do**
    Select $v_{\text{src}} \in V$ with the minimum degree.
    Select $v_{\text{dest}} \in \text{Nei}(v_{\text{src}})$ with the minimum degree.
    Remove edge $e = (v_{\text{src}}, v_{\text{dest}})$ from $E$.
    Append $e$ to $\sigma_E$.
    Update the degrees of $v_{\text{src}}$ and $v_{\text{dest}}$.
**end while** Reverse $\sigma_E$

---

$(v_{\text{src}}^{id}, v_{\text{dest}}^{id}, e^c)$, where the first two elements define the edge connection and $e^c \in \{1, \ldots, K_e\}$ is the edge type. For a graph without node or edge labels, the above representation can be simplified by removing the node or edge labels from the sequence. A graph $G$ with $n$ nodes and $m$ edges can be represented as

$$[\underbrace{v_1^c, v_1^{id}, \ldots, v_n^c, v_n^{id}}_{n \times 2}, a_\Delta, \underbrace{v_{\text{src}}^{id}, v_{\text{dest}}^{id}, e_1^c, \ldots, v_{\text{src}}^{id}, v_{\text{dest}}^{id}, e_m^c}_{m \times 3}].$$

Here $a_\Delta$ is the special token separating node tokens from edge tokens. We illustrate it in Figure 1.

The sequence depends on a node order and an edge order. Nodes are ordered randomly. Nodes in the first part of the sequence follow the node order, and thus $v_i^{id} = i$. The edge order is determined by reversing a degree-based edge-removal process, as described in Alg. 1. The removal process prioritizes the removal of edges connected to low-degree nodes, so its reverse constructs a compact, relatively dense core first, followed by the addition of edges with fewer connections. Edges in the second part of the sequence follow this construction order. We also explore the effectiveness of using other edge orderings such as breadth-first search (BFS) and depth-first search (DFS) (details are presented in Appendix D.1).

### 3.2. Learning Graph Sequences via Transformer

We utilize a Transformer decoder (Vaswani, 2017) for modeling the graph sequences. Before this, we construct a unified vocabulary containing all previously introduced tokens. A tokenizer is defined to map each token to a unique integer.

**Tokenization.** Let $n_{\max}$ be the maximum number of nodes of a graph dataset. The unified vocabulary is then defined as

$$\text{tokenize}(v^{id}) = v^{id}, \quad v^{id} \in \{1, \ldots, n_{\max}\};$$
$$\text{tokenize}(v^c) = v^c + n_{\max}, \quad v^c \in \{1, \ldots, K_v\};$$
$$\text{tokenize}(e^c) = e^c + n_{\max} + K_v, \quad e^c \in \{1, \ldots, K_e\};$$
$$\text{tokenize}(a_\Delta) = n_{\max} + K_v + K_e + 1.$$

We additionally introduce special tokens [SOG] and [EOG], representing the start and the end of the sequence generation.

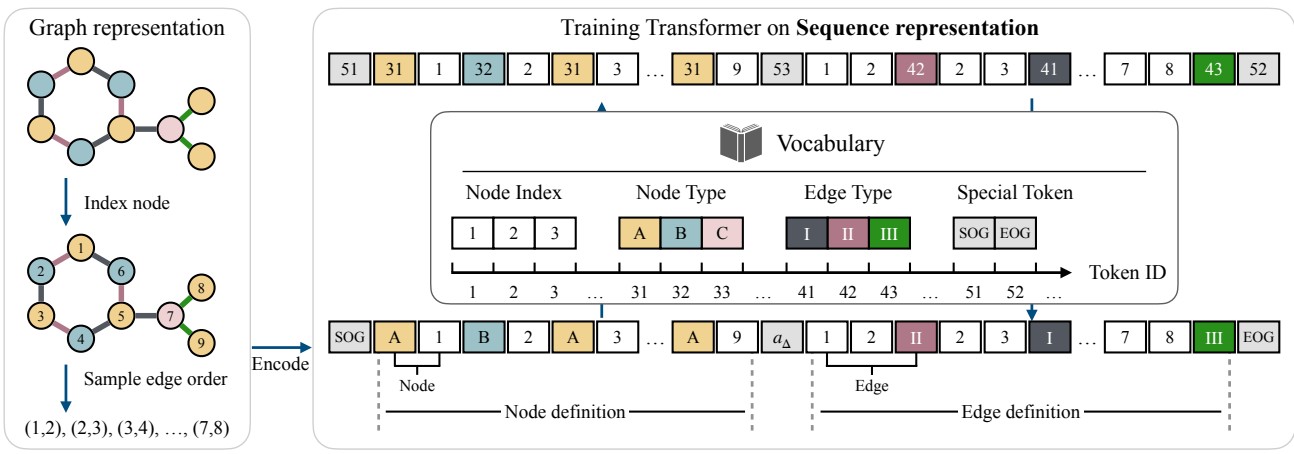

*Figure 1.* Illustration of our proposed graph sequence representation. This representation can be viewed as a sequence of actions: first generating all nodes (node type, node index), then explicitly adding edges (source node index, destination node index, edge type) step by step until completion. A unified vocabulary is used to map different types of actions into a shared token space.

We denote the tokenized sequence $\mathbf{s} = [s_1, \ldots, s_L]$, which is used in the following sections.

**Training objective.** With the autoregressive Transformer, we minimize the negative log-likelihood of the sequence.

$$\mathcal{L}_{\mathrm{pt}}(\mathbf{s}; \theta) := -\log p_\theta(\mathbf{s}) = \sum_{l=1}^{L} -\log p_\theta(s_l | \mathbf{s}_{<l}), \quad (1)$$

where $\theta$ denotes model parameters. With the rules of forming sequences from graphs, not all tokens are legal for $s_\ell$ at a prediction step $p_\theta(s_l | \mathbf{s}_{<l})$. For example, when the current token is a node type, the next token can only be one of the node indices. Illegal tokens can be avoided by masking out their logits in the model output. However, our experiments show that unconstrained logits also yield superior performance thanks to the learning power of Transformers.

**Relationship with graph likelihood.** We show that maximizing the sequence likelihood maximizes a lower bound of the graph likelihood. Denote a training set of $N$ graphs $\{G_1, \ldots, G_N\}$. The log-likelihood of the graph dataset is $\mathbb{L}(\theta) = \sum_{i=1}^{N} \log p_\theta(G_i)$. Let $p_{\mathrm{d}}(G)$ denote the data distribution, and $p_{\mathrm{d}}(\mathbf{s}|G)$ denote the distribution of drawing sequences from a graph $G$. We have

$$\mathbb{L}(\theta) + \mathbb{H}[p_{\mathrm{d}}(G)] = -\mathrm{KL}\big(p_d(G)\|p_\theta(G)\big)$$

$$\geq -\mathrm{KL}\big(p_{\mathrm{d}}(G)\|p_\theta(G)\big) - \mathbb{E}_{p_{\mathrm{d}}}\Big[\mathrm{KL}\big(p_{\mathrm{d}}(\mathbf{s}|G)\|p_\theta(\mathbf{s}|G)\big)\Big]$$

$$= -\mathrm{KL}\big(p_{\mathrm{d}}(\mathbf{s})\|p_\theta(\mathbf{s})\big) = \sum_{i=1}^{N} -\mathcal{L}_{\mathrm{pt}}(\mathbf{s}_i; \theta) + \mathbb{H}[p_{\mathrm{d}}(\mathbf{s})].$$

In the third line, we use the fact that $p(G, \mathbf{s}) = p(\mathbf{s})$ because $\mathbf{s}$ determines $G$. Note that both the entropy terms are constant with respect to the model parameters $\theta$.

From the lower bound, we see that an accurate approximation $p_\theta(\mathbf{s}|G)$ can tighten the bound. Although $p_\theta(\mathbf{s}|G)$ is

not directly compute, providing more sequences from the same graph $G$ can reduce the condition KL. Intuitively, increasing the number of sequences increases the diversity of the training data, thus improving the model's generalization. Combining graph datasets and random sequences, we use the objective (1) to pre-train a Transformer model as the generative model.

## 4. Fine tuning

After pre-training a model, we further fine-tune it for downstream tasks. We consider generative (§4.1) and predictive (§4.2) downstream tasks, where the former aims to generate graphs with desired properties, and the latter utilizes the graph embeddings learned from the Transformer to predict properties.

### 4.1. Goal-oriented Generation

Let $z = \zeta(G)$ denote the function that estimates property $z$ of a graph $G$. In goal-oriented generation, our objective is to train a new model that generates graphs with property values closer to a target value $z^*$ than those produced by the pre-trained model. This problem has a broad application such as drug discovery. In this work, we fine-tune the pre-trained model to improve its ability to generate graphs that better meet the specified property criteria. To this end, we explore two strategies: rejection sampling fine-tuning (RFT) and reinforcement learning (RL).

**Rejection sampling fine-tuning.** This approach fine-tunes the model using its generated samples that have the desired property values. Here we consider the case that the property is a scalar and specify and an acceptance function $m_\omega^{z^*}(G) = \mathbb{1}_{|z^* - \zeta(G)| < \omega}$, where the distance tolerance $\omega$ is a hyperparameter.

---
**Algorithm 2** RFT Dataset Construction
---

**Input:** Model $p_\theta$, acceptance function $m_\omega^{z^*}$, data size $B$.
**Output:** Fine-tuning dataset $\mathcal{D}_\omega^{z^*}$
Initialize $\mathcal{D}_\omega^{z^*} \leftarrow \{\ \}$
**while** $|\mathcal{D}_\omega^{z^*}| \neq B$ **do**
  Generate $G \sim p_\theta$.
  **if** $G$ is valid **and** $m_\omega^{z^*}(G) = 1$ **then**
    Append $G$ to $\mathcal{D}_\omega^{z^*}$.
  **end if**
**end while**

---

---
**Algorithm 3** SBS$^k$ combined with RFT
---

**Input:** Model $p_\theta$, thresholds list $[\omega_1 \ldots, \omega_k]$.
**Output:** Fine-tuned model $p_{\theta_k}$
Set $\theta_0 = \theta$.
**for** $i = 1, \ldots, k$ **do**
  Use $p_{\theta_{i-1}}$ as input model, obtain $\mathcal{D}_{\omega_i}^{z^*} \leftarrow$ Alg. 2.
  Fine-tune $\theta_{i-1}$ on $\mathcal{D}_{\omega_i}^{z^*}$, obtain new parameters $\theta_i$.
**end for**

---

We run the generative model and collect valid graphs that satisfy the acceptance function to construct the fine-tuning dataset $\mathcal{D}_\omega^{y^*} = \{G_b\}_{b=1}^B$. The algorithm is shown in Alg. 2. Note that we expect the learned pre-trained model to have the ability to generate a decent fraction of graphs with the desired property.

RFT becomes inefficient if the pre-trained model can rarely generate acceptable samples. To address this, we run the self-bootstrap (SBS) version of RFT to approach the target distribution in multiple rounds. With $k$ tolerances $\omega_1 > \omega_2 > \ldots > \omega_k = \omega$, we obtain a sequence of fine-tuned models by iteratively constructing fine-tuned datasets using the model trained from the previous tolerance. The SBS algorithm combined with RFT is shown in Alg. 3.

**Reinforcement learning.** Denote a target-relevant reward function $r_{z^*}(G)$, we consider a KL-regularized reinforcement learning problem:

$$\phi^* = \arg\max_\phi \mathbb{E}_{p_{\phi(\mathbf{s})}}\big[r_{z^*}(\mathbf{s}) - \rho_1 \mathrm{KL}\big(p_\phi(\mathbf{s})\|p_\theta(\mathbf{s})\big)\big].$$

Here $r_{z^*}(\mathbf{s}) = r_{z^*}(G)$ as $\mathbf{s}$ uniquely decides $G$. The KL divergence $\mathrm{KL}(\cdot\|\cdot)$ prevents the target model from deviating too much from the pre-trained model.

We choose Proximal Policy Optimization (PPO) (Schulman et al., 2017) to effectively train the target model $p_\phi$ without sacrificing stability. The token-level reward is $r_{z^*}(\mathbf{s})$ only at the last token and zero otherwise:

$$R([\mathbf{s}_{<l}, s_l]) = \begin{cases} 0 & s_l \neq [\mathrm{EOG}] \\ r([\mathbf{s}_{<l}, s_l]) & s_l = [\mathrm{EOG}] \end{cases}.$$

Here $\mathbf{s}_{<l}$ is the state of the $l$-th step in a finite trajectory (sequence). The value function of state $\mathbf{s}_{<l}$ under a model $p$

is the expectation of the undiscounted future return:

$$V^p(\mathbf{s}_{<l}) = \mathbb{E}_{p(\mathbf{s}_{\geq l}|\mathbf{s}_{<l})}\big[r(\mathbf{s})\big].$$

A critic model $V_\psi(\mathbf{s}_{<l})$ is then learned to approximate the true value function $V^p(\mathbf{s}_{<l})$ via minimizing the mean absolute error $\mathcal{L}_{\mathrm{critic}}(\psi)$. We parameterize the critic model using a Transformer with the same architecture as the pre-trained model, except that the logits head is replaced with a value head. The parameters of the critic model are also initialized from the pre-trained model.

We use the clipped surrogate objective $\mathcal{L}_{\mathrm{pg-clip}}(\phi)$ in PPO to optimize the actor model. Moreover, to mitigate possible model degradation, we incorporate the pre-training loss $\mathcal{L}_{\mathrm{pt}}(\phi)$ following Zheng et al. (2023) and Liu et al. (2024).

All terms combined, we minimize the objective:

$$\mathcal{L}_{\mathrm{ppo}}(\phi, \psi; \theta) = \mathcal{L}_{\mathrm{pg-clip}}(\phi) + \rho_2 \mathcal{L}_{\mathrm{critic}}(\psi) + \rho_3 \mathcal{L}_{\mathrm{pt}}(\phi).$$

Here $\rho_1, \rho_2, \rho_3$ are loss coefficients. We provide preliminaries of PPO and details of each loss term in Appendix A.

### 4.2. Property Prediction

In a graph-level learning task, we often need to predict the label $y$ of a graph $G$. With the pre-trained model, we fine-tune it to address the graph-level learning task. After the sequence $\mathbf{s}$ is generated from $G$, we extract the activation $\mathbf{h}$ of the final token $\mathbf{s}_L$ from the last Transformer block as the graph representation. The rationale is that the model must have learned the information of the entire graph in order to expand it with further edges. We replace the token-predicting layer with an MLP to predict $y$ based on $\mathbf{h}$:

$$p(y|\mathbf{s}) = \mathrm{Softmax}(\mathrm{Dropout}(\mathrm{Linear}(\mathbf{h}))).$$

We minimize the cross-entropy loss $-\mathbb{E}_{(G,y)\sim\mathcal{C}} \log p(y|G)$ over a graph classification dataset $\mathcal{C}$ to fine-tune the model. Compared to freezing the whole Transformer during training and only updating parameters of the linear layer, we found that unlocking the latter half of the Transformer blocks significantly enhances the performance.

## 5. Experiments

### 5.1. Setup

**Datasets.** We consider both generative tasks and predictive tasks in our experiments. We use three molecule datasets: QM9 (Wu et al., 2018b), MOSES (Polykovskiy et al., 2020), and GuacaMol (Brown et al., 2019); and four generic graph datasets: Planar, Tree, Lobster, and stochastic block model (SBM), which are widely used to benchmark graph generative models. In predictive tasks, we fine-tune models pre-trained from GuacaMol datasets on molecular properties with the benchmark method MoleculeNet (Wu et al., 2018a). Further details are in Appendix B.4,

| Model | Planar | | | | | | Tree | | | | | |
|---|---|---|---|---|---|---|---|---|---|---|---|---|
| | Deg.↓ | Clus.↓ | Orbit↓ | Spec.↓ | Wavelet↓ | V.U.N.↑ | Deg.↓ | Clus.↓ | Orbit↓ | Spec.↓ | Wavelet↓ | V.U.N.↑ |
| GRAN (Liao et al., 2019) | 7e-4 | 4.3e-2 | 9e-4 | 7.5e-3 | 1.9e-3 | 0 | 1.9e-1 | 8e-3 | 2e-2 | 2.8e-1 | 3.3e-1 | 0 |
| BiGG (Dai et al., 2020) | 7e-4 | 5.7e-2 | 3.7e-2 | 1.1e-2 | 5.2e-3 | 5 | 1.4e-3 | 0.00 | 0.00 | 1.2e-2 | 5.8e-3 | 75 |
| DiGress (Vignac et al., 2022) | 7e-4 | 7.8e-2 | 7.9e-3 | 9.8e-3 | 3.1e-3 | 77.5 | 2e-4 | 0.00 | 0.00 | 1.1e-2 | 4.3e-3 | 90 |
| BwR (Diamant et al., 2023) | 2.3e-2 | 2.6e-1 | 5.5e-1 | 4.4e-2 | 1.3e-1 | 0 | 1.6e-3 | 1.2e-1 | 3e-4 | 4.8e-2 | 3.9e-2 | 0 |
| HSpectre (Bergmeister et al., 2023) | 5e-4 | 6.3e-2 | 1.7e-3 | 7.5e-3 | 1.3e-3 | 95 | 1e-4 | 0.00 | 0.00 | 1.2e-2 | 4.7e-3 | 100 |
| GEEL (Jang et al., 2023) | 1e-3 | 1e-2 | 1e-3 | - | - | 27.5 | 1.5e-3 | 0.00 | 2e-4 | 1.5e-2 | 4.6e-3 | 90 |
| DeFoG (Qin et al., 2024) | 5e-4 | 5e-2 | 6e-4 | 7.2e-3 | 1.4e-3 | 99.5 | 2e-4 | 0.00 | 0.00 | 1.1e-2 | 4.6e-3 | 96.5 |
| G2PT$_{small}$ | 4.7e-3 | 2.4e-3 | 0.00 | 1.6e-2 | 1.4e-2 | 95 | 2e-3 | 0.00 | 0.00 | 7.4e-3 | 3.9e-3 | 99 |
| G2PT$_{base}$ | 1.8e-3 | 4.7e-3 | 0.00 | 8.1e-3 | 5.1e-3 | 100 | 4.3e-3 | 0.00 | 1e-4 | 7.3e-3 | 5.7e-3 | 99 |

| Model | Lobster | | | | | | SBM | | | | | |
|---|---|---|---|---|---|---|---|---|---|---|---|---|
| | Deg.↓ | Clus.↓ | Orbit↓ | Spec.↓ | Wavelet↓ | V.U.N.↑ | Deg.↓ | Clus.↓ | Orbit↓ | Spec.↓ | Wavelet↓ | V.U.N.↑ |
| GRAN (Liao et al., 2019) | 3.8e-2 | 0.00 | 1e-3 | 2.7e-2 | - | - | 1.1e-2 | 5.5e-2 | 5.4e-2 | 5.4e-3 | 2.1e-2 | 25 |
| BiGG (Dai et al., 2020) | 0.00 | 0.00 | 0.00 | 9e-3 | - | - | 1.2e-3 | 6.0e-2 | 6.7e-2 | 5.9e-3 | 3.7e-2 | 10 |
| DiGress (Vignac et al., 2022) | 2.1e-2 | 0.00 | 4e-3 | - | - | - | 1.8e-3 | 4.9e-2 | 4.2e-2 | 4.5e-3 | 1.4e-3 | 60 |
| BwR (Diamant et al., 2023) | 3.2e-1 | 0.00 | 2.5e-1 | - | - | - | 4.8e-2 | 6.4e-2 | 1.1e-1 | 1.7e-2 | 8.9e-2 | 7.5 |
| HSpectre (Bergmeister et al., 2023) | - | - | - | - | - | - | 1.2e-2 | 5.2e-2 | 6.7e-2 | 6.7e-3 | 2.2e-2 | 45 |
| GEEL (Jang et al., 2023) | 2e-3 | 0.00 | 1e-3 | - | - | 72.7 | 2.5e-2 | 3e-3 | 2.6e-2 | - | - | 42.5 |
| DeFoG (Qin et al., 2024) | - | - | - | - | - | - | 6e-4 | 5.2e-2 | 5.6e-2 | 5.4e-3 | 8e-3 | 90 |
| G2PT$_{small}$ | 2e-3 | 0.00 | 0.00 | 5e-3 | 8.5e-3 | 100 | 3.5e-3 | 1.2e-2 | 7e-4 | 7.6e-3 | 9.8e-3 | 100 |
| G2PT$_{base}$ | 1e-3 | 0.00 | 0.00 | 4e-3 | 1e-2 | 100 | 4.2e-3 | 5.3e-3 | 3e-4 | 6.1e-3 | 6.9e-3 | 100 |

*Table 2.* Generative performance on generic graph datasets.

**Model specifications.** We train Transformers with three different sizes: (1) the small Transformer has 6 layers and 6 attention heads, with $d_{model} = 384$, leading to approximately 10M parameters; (2) the base Transformer has 12 layers and 12 attention heads, with $d_{model} = 768$, leading to approximately 85M parameters; (2) the large Transformer has 24 layers and 16 attention heads, with $d_{model} = 1024$, leading to approximately 300M parameters. We use different specifications for different experiments according to the task complexity.

## 5.2. A Case Study with Planar Graphs

We compare our token-based representation against the adjacency matrix and validate its effectiveness in the generation task. We train Transformer decoders on planar graphs using our token-based representation and adjacency matrices, and then evaluate their generative performance. For the adjacency representation, planar graphs are encoded as sequences of 0s and 1s derived from the strictly lower triangular matrix, with rows and columns permuted using BFS orderings to augment the training dataset. Table 3 presents the quantitative and qualitative results of the generated samples. Our proposed representation demonstrates superior generative performance with a much smaller set of tokens. In contrast, the model trained with adjacency matrices struggles to capture the topological rule of planar graphs.

## 5.3. Generic Graph Generation

We evaluate G2PT on four generic datasets using Maximum Mean Discrepancy (MMD) to compare the graph statistics distributions of generated and test graphs. The evaluation

| Rep. | #Tokens↓ | Deg.↓ | Clus.↓ | Orbit↓ | Spec.↓ | Wavelet↓ | V.U.N.↑ |
|---|---|---|---|---|---|---|---|
| **A** | 2018 | 8.6e-3 | 1e-1 | 8e-3 | 3.2e-2 | 6.1e-2 | 94 |
| **s** (Ours) | **737** | **4.7e-3** | **2.4e-3** | **0.00** | **1.6e-2** | **1.4e-2** | **95** |

| **A** | | **s** (Ours) | |
|---|---|---|---|

*Table 3.* Generative performance comparison between the proposed edge sequence and adjacency matrix representations.

considers degree (Deg.), clustering coefficient (Clus.), orbit counts (Orbit), spectral properties (Spec.), and wavelet statistics. Moreover, we report the percentage of valid, unique, and novel samples (V.U.N.) (Vignac et al., 2022). For this task, we trained the G2PT$_{small}$ and G2PT$_{base}$ models.

As shown in Table 2, G2PT demonstrates superior performance compared to the baselines. The details about baseline and metric are introduced in appendix B.5 The base model achieves 11 out of 24 best scores and ranks in the top two for 17 out of 24 metrics. The small model also demonstrates competitive results, indicating that a lightweight model can effectively capture the graph patterns in the datasets.

## 5.4. Molecule Generation

De novo molecular design is a key real-world application of graph generation. We assess G2PT's performance on the QM9, MOSES, and GuacaMol datasets. For the QM9 dataset, we adopt the evaluation protocol in Vignac et al. (2022). For MOSES and GuacaMol, we utilize the evaluation pipelines provided by their respective toolkits (Polykovskiy et al., 2020; Brown et al., 2019).

| Model | MOSES | | | | | | | GuacaMol | | | | |
|---|---|---|---|---|---|---|---|---|---|---|---|---|
| | Validity↑ | Unique.↑ | Novelty↑ | Filters↑ | FCD↓ | SNN↑ | Scaf↑ | Validity↑ | Unique.↑ | Novelty↑ | KL Div.↑ | FCD↑ |
| LigGPT (Bagal & Aggarwal) | 90.0 | 99.9 | 94.1 | - | - | - | - | 98.6 | 99.8 | **100** | - | - |
| DiGress (Vignac et al., 2022) | 85.7 | **100** | 95.0 | 97.1 | 1.19 | 0.52 | 14.8 | 85.2 | **100** | 99.9 | 92.9 | 68 |
| GEEL (Jang et al., 2023) | 92.1 | **100** | 81.1 | 97.5 | 1.28 | 0.52 | 3.6 | 88.2 | 98.2 | 89.1 | 93.1 | 71.5 |
| DisCo (Xu et al., 2024) | 88.3 | **100** | 97.7 | 95.6 | 1.44 | 0.5 | 15.1 | 86.6 | 86.6 | 86.5 | 92.6 | 59.7 |
| Cometh (Siraudin et al., 2024) | 90.5 | 99.9 | 92.6 | 99.1 | 1.27 | 0.54 | **16.0** | 98.9 | 98.9 | 97.6 | 96.7 | 72.7 |
| DeFoG (Qin et al., 2024) | 92.8 | 99.9 | 92.1 | **99.9** | 1.95 | **0.55** | 14.4 | **99.0** | 99.0 | 97.9 | **97.9** | 73.8 |
| G2PT$_{small}$ | 95.1 | **100** | 91.7 | 97.4 | 1.10 | 0.52 | 5.0 | 90.4 | **100** | 99.8 | 92.8 | 86.6 |
| G2PT$_{base}$ | 96.4 | **100** | 86.0 | 98.3 | **0.97** | **0.55** | 3.3 | 94.6 | **100** | 99.5 | 96.0 | **93.4** |
| G2PT$_{large}$ | **97.2** | **100** | 79.4 | 98.9 | 1.02 | **0.55** | 2.9 | 95.3 | **100** | 99.5 | 95.6 | 92.7 |

| Model | QM9 | | | MOSES | | | GuacaMol | | |
|---|---|---|---|---|---|---|---|---|---|
| | Validity↑ | Unique.↑ | FCD↓ | Train | G2PT$_{small}$ | G2PT$_{base}$ | Train | G2PT$_{small}$ | G2PT$_{base}$ |
| DiGress (Vignac et al., 2022) | 99.0 | 96.2 | - | | | | | | |
| DisCo (Xu et al., 2024) | **99.6** | 96.2 | 0.25 | | | | | | |
| Cometh (Siraudin et al., 2024) | 99.2 | 96.7 | 0.11 | | | | | | |
| DeFoG (Qin et al., 2024) | 99.3 | 96.3 | 0.12 | | | | | | |
| G2PT$_{small}$ | 99.0 | 96.7 | **0.06** | | | | | | |
| G2PT$_{base}$ | 99.0 | **96.8** | **0.06** | | | | | | |
| G2PT$_{large}$ | 98.9 | 96.7 | **0.06** | | | | | | |

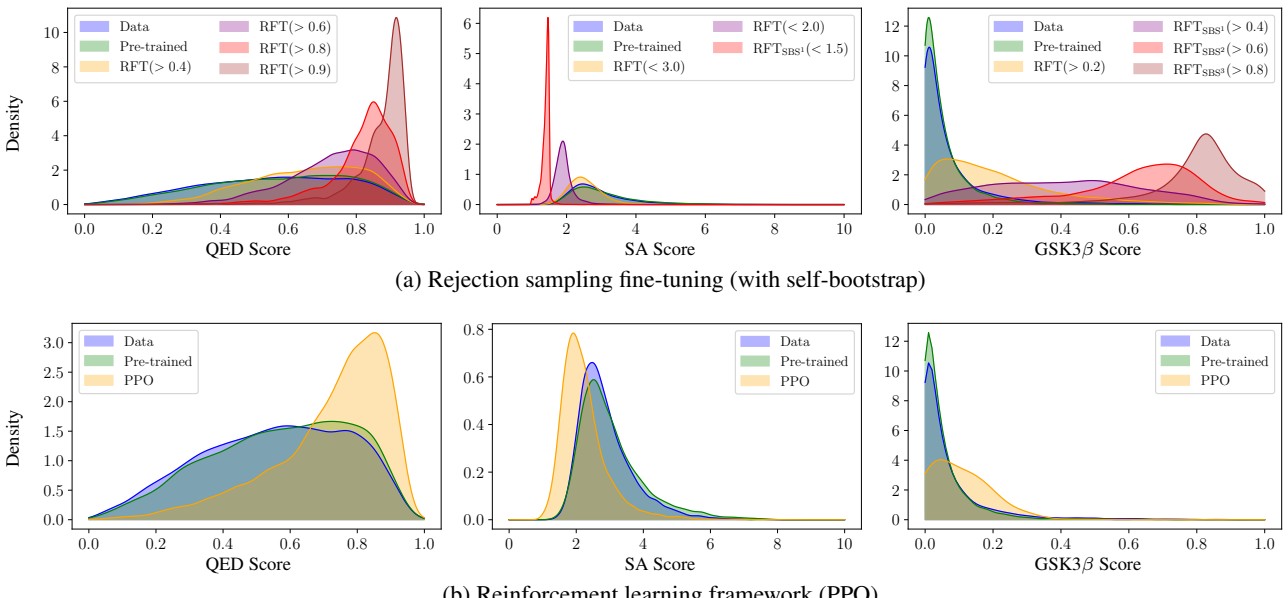

*Table 4.* Generative performance on molecular graph datasets

*Figure 2.* Goal-oriented molecule generation using QED, SA and GSK3$\beta$ scores. Top row (a) shows the results using RFT, and bottom row (b) shows the results using RL.

The quantitative results are presented in Table 4. On MOSES, G2PT surpasses other state-of-the-art models in validity, uniqueness, FCD, and SNN metrics. We introduce the details for metrics in appendix B.6. Notably, the FCD, SNN, and scaffold similarity (Scaf) evaluations compare generated samples to a held-out test set, where the test molecules have scaffolds distinct from the training data. Although the scaffold similarity score is relatively low, the overall performance indicates that G2PT achieves a better goodness of fit on the training set. G2PT also delivers strong performance on the GuacaMol and QM9 datasets. We additionally provide qualitative examples from the MOSES and GuacaMol datasets in the table.

## 5.5. Goal-oriented Generation

In addition to distribution learning which aims to draw independent samples from the learned graph distribution, goal-oriented generation is a major task in graph generation that aims to draw samples with additional constraints or preferences and is key to many applications such as molecule optimization (Du et al., 2024).

We validate the capability of G2PT on goal-oriented generation by fine-tuning the pre-trained model. Practically,

| | BBBP | Tox21 | ToxCast | SIDER | ClinTox | MUV | HIV | BACE | Avg. |
|---|---|---|---|---|---|---|---|---|---|
| AttrMask (Hu et al., 2020a) | 70.2±0.5 | 74.2±0.8 | 62.5±0.4 | 60.4±0.6 | 68.6±9.6 | 73.9±1.3 | 74.3±1.3 | 77.2±1.4 | 70.2 |
| InfoGraph (Sun et al., 2020) | 69.2±0.8 | 73.0±0.7 | 62.0±0.3 | 59.2±0.2 | 75.1±5.0 | 74.0±1.5 | 74.5±1.8 | 73.9±2.5 | 70.1 |
| ContextPred (Hu et al., 2020a) | **71.2±0.9** | 73.3±0.5 | 62.8±0.3 | 59.3±1.4 | 73.7±4.0 | 72.5±2.2 | 75.8±1.1 | 78.6±1.4 | 70.9 |
| GraphCL (You et al., 2021) | 67.5±2.5 | **75.0±0.5** | 62.8±0.2 | 60.1±1.3 | 78.9±4.2 | **77.1±1.0** | 75.0±0.4 | 68.7±7.8 | 70.6 |
| GraphMVP (Liu et al., 2022a) | 68.5±0.2 | 74.5±0.0 | 62.7±0.1 | **62.3±1.6** | 79.0±2.5 | 75.0±1.4 | 74.8±1.4 | 76.8±1.1 | 71.7 |
| GraphMAE (Hou et al., 2022b) | 70.9±0.9 | **75.0±0.4** | **64.1±0.1** | 59.9±0.5 | 81.5±2.8 | 76.9±2.6 | **76.7±0.9** | 81.4±1.4 | **73.3** |
| G2PT$_{small}$ (No pre-training) | 60.7±0.3 | 66.4±0.5 | 57.0±0.3 | 61.6±0.2 | 67.8±1.1 | 45.8±8.5 | 70.1±7.5 | 68.8±1.3 | 62.3 |
| G2PT$_{base}$ (No pre-training) | 56.5±0.2 | 67.4±0.4 | 57.9±0.1 | 60.2±2.8 | 71.0±5.6 | 60.1±1.3 | 72.7±1.1 | 73.4±0.3 | 64.9 |
| G2PT$_{small}$ | 68.5±0.5 | 74.7±0.2 | 61.2±0.1 | 61.7±1.0 | **82.3±2.2** | 74.9±0.1 | 75.7±0.4 | 81.3±0.5 | 72.5 |
| G2PT$_{base}$ | 71.0±0.4 | **75.0±0.3** | 63.0±0.5 | 61.9±0.2 | 82.1±1.1 | 74.5±0.3 | 76.3±0.4 | **82.3±1.6** | **73.3** |

*Table 5.* Results for molecule property prediction in terms of ROC-AUC. We report mean and standard deviation over three runs.

we employ the model pre-trained on GuacaMol dataset and select three commonly used physiochemical and binding-related properties: quantitative evaluation of druglikeness (QED), synthesis accessibility (SA), and the activity against target protein Glycogen synthase kinase 3 beta (GSK3$\beta$), detailed in Appendix B.3. The property oracle functions are provided by the Therapeutics Data Commons (TDC) package (Huang et al., 2022).

As discussed in §4.1, we employ two approaches for fine-tuning: (1) rejection sampling fine-tuning and (2) reinforcement learning with PPO. Figure 2 shows that both methods can effectively push the learned distribution to the distribution of interest. Notably, RFT, with up to three rounds of SBS, significantly shifts the distribution towards a desired one. In contrast, PPO, despite biasing the distribution, suffers from the over-regularization from the base policy, which aims for training stability. In the most challenging case (GSK3$\beta$), PPO fails to sampling data with high rewards. Conversely, RFT overcomes the barrier in the second round (RFT$_{SBS^1}$), where its distribution becomes flat across the range and quickly transitions to a high-reward distribution.

### 5.6. Predictive Performance on Downstream Tasks

We conduct experiments on eight graph classification benchmark datasets from MoleculeNet (Wu et al., 2018a), strictly following the data splitting protocol used in GraphMAE (Hou et al., 2022a) for fair comparison. A detailed description of these datasets is provided in Appendix B.4.

For downstream fine-tuning, we initialize G2PT with parameters pre-trained on the GuacaMol dataset, which contains molecules with up to 89 heavy atoms. We also provide results where models are not pre-trained.

As summarized in Table 5, G2PT's graph embeddings demonstrate consistently strong (best or second-best) performance on seven out of eight downstream tasks, achieving an overall performance comparable to GraphMAE, a leading self-supervised learning (SSL) method. Notably, while previous SSL approaches leverage additional features such

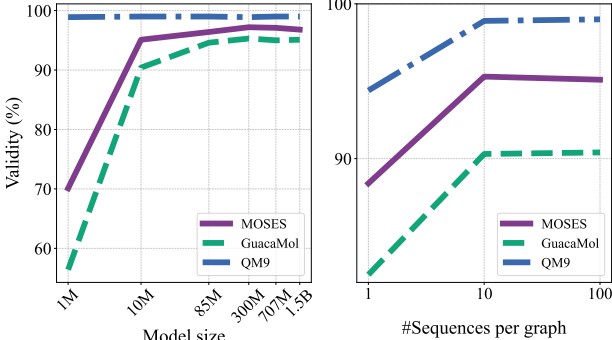

*Figure 3.* Model and data scaling effects.

as 3D information or chirality, G2PT is trained exclusively on 2D graph structural information. Overall, these results indicate that G2PT not only excels in generation but also learns effective graph representations.

### 5.7. Scaling Effects

We analyze how scaling the model size and data size will affect the model performance using the three molecular datasets. We use the validity score to quantify the model performance. Results are provided in Figure 3.

For model scaling, we additionally train G2PTs with 1M, 707M, and 1.5B parameters. We notice that as model size increases, validity score generally increases and saturates at some point, depending on the task complexity. For instance, QM9 saturates at the beginning (1M parameters) while MOSES and GuacaMol require more than 85M (base) parameters to achieve satisfying performance.

For data scaling, we generating multiple sequences from the same graph to improve the diversity of the training data. The number of augmentation per graph is chosen from $\{1, 10, 100\}$. As shown, one sequence per graph is insufficient to train Transformers effectively, and improving data diversity helps improve model performance. Similar to model scaling, performance saturated at some point when enough data are used.

# 6. Conclusion

This work revisits the sequential approach to graph generation and proposes a novel token-based representation that efficiently encodes graph structures via node and edge tokens. This representation serves as the foundation for the proposed Graph Generative Pre-trained Transformer (G2PT), an auto-regressive model that effectively models graph sequences using next-token prediction. Extensive evaluations demonstrated that G2PT achieves remarkable performance across multiple datasets and tasks, including generic graph and molecule generation, as well as downstream tasks like goal-oriented graph generation and graph property prediction. The results highlight G2PT's adaptability and scalability, making it a versatile framework for various applications. One limitation of our method is that G2PT is order-sensitive, where different graph domains may prefer different edge orderings. Future research could be done by exploring edge orderings that are more universal and expressive.

# Impact Statement

This paper introduces a framework that models graphs in a similar vein to GPT (Generative Pre-trained Transformer). The G2PT framework allows seamless plantation of training techniques that have developed in other domains based on GPT. Besides performing generative tasks such as drug discovery, G2PT also can be easily extended for discriminative tasks such as graph property prediction. We hope this work will advance the field of graph learning. As a powerful tool, G2PT may also be used as one step in a complex system to create molecule structures harmful to humans or the environment, but we don't see immediate hazards from our study.

# Acknowledgment

We thank all reviewers for their insightful feedback. Chen and Liu's work was supported by NSF 2239869. He and Xu's work was support by NSF 2239672.

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

# A. Reinforcement Learning Details

## A.1. Preliminaries on Proximal Policy Optimization (PPO)

**Generalized Advantage Estimation.** In reinforcement learning, the Q function $Q(\mathbf{s}_{<l}, s_l)$ captures the expected returns when taking an action $s_l$ at current state $\mathbf{s}_{<l}$, and the value function $V(\mathbf{s}_{<l})$ captures the expected return following the policy from a given state $\mathbf{s}_{<l}$.

The advantage function $A(s_l, \mathbf{s}_{<l})$, defined as the difference between the Q function and the value function, measures whether taking action $s_l$ is better or worse than the policy's default behavior. In practice, the Q function is estimated using the actual rewards $r_l$ and the estimated future returns (the value function). There are two commonly used estimators, one is the one-step Temporal Difference (TD):

$$\hat{Q}(\mathbf{s}_{<l}, s_l) = r_l + \gamma V(\mathbf{s}_{<l+1}),$$
$$\hat{A}(\mathbf{s}_{<l}, s_l) = r_l + \gamma V(\mathbf{s}_{<l+1}) - V(\mathbf{s}_{<l}),$$

and the full Monte Carlo (MC):

$$\hat{Q}(\mathbf{s}_{<l}, s_l) = \sum_{l'=l}^{L} \gamma^{l'-l} r_{l'},$$
$$\hat{A}(\mathbf{s}_{<l}, s_l) = \sum_{l'=l}^{L} \gamma^{l'-l} r_{l'} - V(\mathbf{s}_{<l}),$$

assuming finite trajectory with $L$ steps in total. However, the TD estimator exhibits high bias and the MC estimator exhibits high variance. The Generalized Advantage Estimation (GAE) (Schulman et al., 2015) effectively balances the high bias and high variance smoothly using a trade-off parameter $\gamma$. Let $\delta_l = r_t + \gamma V(\mathbf{s}_{<l+1}) - V(\mathbf{s}_{<l})$, the definition of GAE is:

$$\hat{A}^{\gamma}(\mathbf{s}_{<l}, s_l) = \sum_{l'=l}^{L} (\gamma\lambda)^{l'-l} \delta_{l'} = \delta_l + \hat{A}^{\gamma}(\mathbf{s}_{<l+1}, s_{l+1}).$$

GAE plays an important role in estimating the policy gradient, and will be used in the PPO algorithm.

**Proximal Policy Optimization.** PPO (Schulman et al., 2017) is a fundamental technique in reinforcement learning, designed to train policies efficiently while preserving stability. It is built on the principle that gradually guides the policy towards an optimal solution, rather than applying aggressive updates that could compromise the stability of the learning process.

In traditional policy gradient methods, the new policy should remain close to the old policy in the parameters space. However, proximity in parameter space does not indicate similar performance. A large update step in policy may lead to falling "off the cliff", thus getting a bad policy. Once it is stuck in a bad policy, it will take a very long time to recover.

PPO introduces two kinds of constraints on policy updates. The first kind is to add an KL-regularization term to the policy gradient "surrogate" objective

$$\mathcal{L}_{\text{pg-penalty}}(\phi) = \hat{\mathbb{E}}_l \left[ \frac{p_\phi(s_l|\mathbf{s}_{<l})}{p_{\phi_{\text{old}}}(s_l|\mathbf{s}_{<l})} \hat{A}_l \right] - \beta \text{KL}(p_{\phi_{\text{old}}}(\cdot|\mathbf{s}_{<l}) \| p_\phi(\cdot|\mathbf{s}_{<l})).$$

Here $\hat{\mathbb{E}}_l[\cdot]$ is the empirical average over a finite batch of samples where sampling and optimization alternates. $\beta$ is the penalty factor. $\hat{A}_l := \hat{A}^{\gamma}(\mathbf{s}_{<l}, s_l)$ is GAE, which is detailed in last section.

The second type is the **clipped surrogate objective**, expressed as

$$\mathcal{L}_{\text{pg-clip}}(\phi) = \hat{\mathbb{E}}_l \left[ \min \left( \frac{p_\phi(s_l|\mathbf{s}_{<l})}{p_{\phi_{\text{old}}}(s_l|\mathbf{s}_{<l})} \hat{A}_l, \text{clip}\left( \frac{p_\phi(s_l|\mathbf{s}_{<l})}{p_{\phi_{\text{old}}}(s_l|\mathbf{s}_{<l})}, 1 - \epsilon, 1 + \epsilon \right) \hat{A}_l \right) \right],$$

where $\frac{p_\phi(s_l|\mathbf{s}_{<l})}{p_{\phi_{\text{old}}}(s_l|\mathbf{s}_{<l})}$ is the probability ratio between the new and the old policy. And $\epsilon$ decides how much the new policy can deviate from the one policy. The clipping operations prevent the policy from changing too much from the older one within one iteration. In the following, we elaborate on how the critic model is optimized.

| | QM9 | MOSES | GuacaMol | Planar | Tree | Lobster | SBM |
|---|---|---|---|---|---|---|---|
| #Node Types | 4 | 8 | 12 | 1 | 1 | 1 | 1 |
| #Edge Types | 4 | 4 | 4 | 1 | 1 | 1 | 1 |
| Avg. #Nodes | 8.79 | 21.67 | 27.83 | 64 | 64 | 55 | 104.01 |
| Min. #Nodes | 1 | 8 | 2 | 64 | 64 | 10 | 44 |
| Max. #Nodes | 9 | 27 | 88 | 64 | 64 | 100 | 187 |
| #Training Sequences | 9,773,200 | 141,951,200 | 111,863,300 | 12,800,000 | 10,000,000 | 12,800,000 | 12,800,000 |
| Vocabulary Size | 27 | 60 | 120 | 73 | 73 | 110 | 195 |
| Max Sequence Length | 85 | 207 | 614 | 737 | 383 | 599 | 3950 |

*Table 6.* Dataset statistics.

**Value Function Approximation.** The critic model $V_\psi(\mathbf{s}_{<l})$ in PPO algorithm is used to approximate the actual value function $V^p(\mathbf{s}_{<l})$. We use the mean absolute value loss to minimize the difference between the predicted values and the actual return values. Specifically, the objective is

$$\mathcal{L}_{\text{critic}}(\psi) = \hat{\mathbb{E}}_l\big[|V_\psi(\mathbf{s}_{<l}) - \hat{V}(\mathbf{s}_{<l})|\big].$$

Here the actual return value is estimated using GAE to balance the bias and variance:

$$\hat{V}(\mathbf{s}_{<l}) = \hat{A}(\mathbf{s}_{<l}, s_l) + V_{\psi_{\text{old}}}(\mathbf{s}_{<l}),$$

where $V_{\psi_{\text{old}}}(\mathbf{s}_{<l})$ is collected during the sampling step in PPO. The critic loss is weight by a factor $\rho_2$.

### A.2. KL-regularization

As mentioned in §4.1, we adopt a KL-regularized reinforcement learning approach. Unlike the KL penalty in $\mathcal{L}_{\text{pg-penalty}}(\phi)$, this regularizer ensures that the policy model $p_\phi$ does not diverge significantly from the reference model $p_\theta$. Instead of optimizing this term directly, we incorporate it into the rewards $r_l$. Specifically, we define:

$$r_l^{\rho_1} = r_l - \rho_1 \text{KL}(p_\phi(\cdot|[\mathbf{s}_{<l}, s_l])\|p_\theta(\cdot|[\mathbf{s}_{<l}, s_l])),$$

where $\rho_1$ is the penalty factor. In practice, $\rho_1$ is set to a small value, such as 0.03, to promote exploration.

### A.3. Pre-training loss

Following Zheng et al. (2023) and Liu et al. (2024), we incorporate the pre-training loss $\mathcal{L}_{\text{pt}}(\phi)$ o mitigate potential degeneration in the model's ability to produce valid sequences. This is particularly beneficial for helping the actor model recover when it "falls off the grid" during PPO. The pre-training data is sourced from the dataset used to train the reference model, and the loss $\mathcal{L}_{\text{pt}}(\phi)$ is weighted by the coefficient $\rho_3$.

## B. Additional Experimental Details

### B.1. Graph Generative Pre-training

Generative pre-training leverages graph-structured data to learn foundational representations that can be fine-tuned for downstream tasks.

**Sequence conversion.** We convert graphs into sequences of tokens that represent nodes and edges. This transformation involves encoding the molecular structure in a sequential format that captures both the composition and the order of assembly. For instance, we iteratively process the nodes and edges, and insert special tokens to mark key points in the sequence, such as the start and end of generation. Additionally, we apply preprocessing steps like filtering molecules by size, removing hydrogens, or addressing dataset-specific constraints to ensure consistency and suitability for the target tasks.

**Data splitting.** We divide generic datasets into training, validation, and test sets based on splitting ratios 6:2:2. For the molecular datasets, we follow the default settings of the datasets.

| | 10M | 85M | 300M |
|---|---|---|---|
| Architecture | | | |
| #layers | 6 | 12 | 24 |
| #heads | 6 | 12 | 16 |
| $d_{\text{model}}$ | 384 | 768 | 1024 |
| dropout rate | | 0.0 | |
| Training | | | |
| Lr | | 1e-4 | |
| Optimizer | | AdamW (Loshchilov & Hutter, 2019) | |
| Lr scheduler | | Cosine | |
| Weight decay | | 1e-1 | |
| #iterations | | 300000 | |
| Batch size | 60 | 60 | 30 |
| #Gradient Accumulation | 8 | 8 | 16 |
| Grad Clipping Value | | 1 | |
| #Warmup Iterations | | 2000 | |

*Table 7.* Hyperparameters for graph generative pre-training.

**Dataset statistics.** The vocabulary size, maximum sequence length, and other parameters vary across datasets due to their distinct molecular characteristics. We summarize the specifications in Table 6, which includes details on the number of node types, edge types, and graphs for each dataset.

**Hyperparameters.** Table 7 provides hyperparameters used for training three distinct model sizes, corresponding to approximately 10M, 85M, and 300M parameters, respectively.

### B.2. Demonstration Experiment

We elaborate on how to represent adjacency matrix as sequence and train a transformer decoder on it. We choose planar graphs as the investigation object as it requires a model to be able to capture the rule embedded in the graph. We use G2PT$_{\text{small}}$ for this experiment.

**Sequence conversion.** We convert a 2-D adjacency matrix into a 1-D sequence before training the models. Similar to GraphRNN (You et al., 2018b), we consider modeling the strictly lower triangle of the adjacency matrix. To obtain sequence, we flatten the triangle by concatenating the rows together. The $i$-th row has $i - 1$ entries, where each entry is either 0 or 1. We employ BFS to determine the node orderings, which is used to permute the rows and columns of the adjacency matrix to reduce the learning complexity (as uniform orderings are generally harder to fit (Chen et al., 2021)).

**Training transformers on adjacency matrices.** After obtaining the sequence representation, we prepend and append two special tokens, [SOG] and [EOG], to mark the start and end of the generation of each sequence. The sequence is then tokenized using a vocabulary of size 4, and the transformer is trained on these sequences. Note that no additional token is needed to indicate transitions between rows, as the flattened sequence maintains a fixed correspondence between positions and the referenced node pairs. Specifically, the original row and column indices in the adjacency matrix for the $i$-th entry in the sequence can be determined as:

$$\text{row} = \left\lceil \frac{1 + \sqrt{1 + 8i}}{2} \right\rceil, \text{col} = i - \frac{(\text{row} - 1)(\text{row} - 2)}{2}.$$

Here $\lceil \cdot \rceil$ is the ceiling operation. Such correspondence is agnostic to graph size and can be inferred by transformers by using positional embeddings.

|  | QED | SA | GSK3$\beta$ |
|---|---|---|---|
| $\gamma$ |  | 1.0 |  |
| $\lambda$ |  | 1.0 |  |
| $\rho_1$ |  | 0.5 |  |
| $\rho_2$ | 0.03 | 0.03 | 0.05 |
| $\rho_3$ |  | 0.03 |  |
| Advantage Normalization and Clipping | Yes | No | No |
| Reward Normalization and Clipping | No | Yes | Yes |
| Ratio Clipping ($\epsilon$) |  | [0.2] |  |
| Critic Value Clipping |  | [0.2] |  |
| Entropy Regularization |  | No |  |
| Gradient Clipping Value |  | 1.0 |  |
| Actor Lr |  | 1.0 |  |
| Critic Lr | 0.5 | 0.5 | 1.0 |
| #Iterations |  | 6000 |  |
| Batch size |  | 60 |  |

*Table 8.* Hyperparameters used for PPO training.

### B.3. Fine-tuning G2PT for Goal-oriented Generation

For the goal-oriented generation, we fine-tune G2PT to generate molecules with desired characteristics. Specifically, we consider three properties that are commonly used for molecule optimization whose functions are easily accessed through the Therapeutics Data Commons (TDC) package (Huang et al., 2022).

- Quantitative evaluation of druglikeness (QED): range 0-1, the higher the more druglike.

- Synthesis accessibility (SA) score: range 1-10, the lower the more synthesizable.

- GSK3$\beta$: activity against target protein Glycogen synthase kinase 3 beta, range 0-1, the higher the better activity.

We use the 85M model pre-trained on GucaMol dataset for all experiments. Below we elaborate on how the RFT and RL algorithms implement each optimization task (property).

**Rejection-sampling fine-tuning.** For RFT algorithm without SBS, we begin by generating samples using the pre-trained model and retain only those that meet the criteria defined by the acceptance function $m_\omega^{z^*}(\cdot)$. We collect 200,000 qualified samples from the generations. Then, we fine-tune the model by initializing it with pre-trained parameters. When combining RFT with SBS, we repeat this process iteratively, using the fine-tuned model from the previous iteration for both sampling and parameter initialization.

For QED score, we retain samples with scores exceeding thresholds of 0.4, 0.6, 0.8, or 0.9. We do not use the SBS algorithm here, as the pre-trained model generates samples efficiently across all QED score ranges.

For SA score, we consider thresholds of $\{< 3.0, < 2.0, < 1.5\}$. We find that the pre-trained model efficiently generates molecules with SA scores below 2.0 and 3.0 but struggles with scores below 1.5. To address this, we bootstrap the fine-tuned model from the 2.0 threshold to the 1.5 threshold.

For GSK3$\beta$, we consider thresholds in $\{> 0.2, > 0.4, > 0.6, > 0.8\}$. We observe that the pre-trained model's score distribution is skewed towards 0, making it challenging to generate satisfactory samples. To resolve this, we fine-tune the model at the 0.2 threshold and progressively bootstrap it through intermediate thresholds (0.4, 0.6) up to 0.8, performing three bootstrapping steps in total.

All models are trained for 6000 iterations, with batch size of 120 and learning rate of 1e-5. The learning rate gradually decay to 0 using Cosine scheduler.

**Reinforcement learning.** We use the PPO algorithm to further optimize the pre-trained model. In practice, the token-level reward $R([\mathbf{s}_{<l}, s_l])$ is set to 0 except when $s_l = [\text{EOG}]$. The final reward $r(\mathbf{s})$ for the three properties are designed as follow:

$$r^{\text{QED}}(\mathbf{s}) = \mathbb{1}_{s \to G} \max(0.2, 2 \times (\text{QED}(G) - 0.5)), \tag{2}$$

$$r^{\text{SA}}(\mathbf{s}) = \mathbb{1}_{s \to G} \max(0.2, 0.2 \times (5 - \text{SA}(G)), \tag{3}$$

$$r^{\text{GSK3}\beta}(\mathbf{s}) = \mathbb{1}_{s \to G}(5 \times (\text{GSK3}\beta(G)). \tag{4}$$

The indicator function $\mathbb{1}_{s \to G}$ assigns 0 to the final reward when the generated sequence $\mathbf{s}$ is invalid. We show the PPO hyperparameters for each targeted task in Table 8.

### B.4. Fine-tuning G2PT for Graph Property Prediction

**Datasets.** We use eight classification tasks in MoleculeNet (Wu et al., 2018a) following Zhu et al. (2024) to validate the predictive capability of our learned representations.

The datasets cover two types of molecular properties: biophysical and physiological properties.

- Biophysical properties include (1) the HIV dataset for HIV replication inhibition, (2) the Maximum Unbiased Validation (MUV) dataset for virtual screening with nearst neighbor search, (3) the BACE dataset for inhibition of human $\beta$-secretase 1 (BACE-1), and (4) the Side Effect Resource (SIDER) dataset for grouping the side effects of marketed drugs into 27 system organ classes.

- Physiological properties include (1) the Blood-brain barrier penetration (BBBP) dataset for predicting barrier permeability of molecules targeting central nervous system, (2) the Tox21, (3) the ClinTox, and (4) the ToxCast datasets that are all associated with certain type of toxicity of the chemical compounds.

We adopt the scaffold split that divides train, validation and test set by different scaffolds, introduced by Wu et al. (2018b).

**Fine-tuning details.** We fine-tune G2PT$_{\text{small}}$ and G2PT$_{\text{base}}$ pre-trained on GuacaMol dataset for the downstream tasks. We setup the dropout rate to 0.5 and use a learning rate of 1e-4 for training the linear layer. For the half transformer blocks, we use a learning rate of 1e-6. We use a batch size of 256 and train the models for 100 epochs. Test result with best validation performance is reported.

### B.5. Baselines

We evaluate our proposed method against a variety of baselines across different datasets. The baselines include models that span diverse methodologies, ranging from graph neural networks to transformer-based architectures.

**Generic graph datasets.** The performance of baseline models on Planar, Tree, Lobster, and SBM datasets is shown in Table 2. We consider baselines mainly from two categories: auto-regressive and diffusion graph models. Among them, GRAN (Liao et al., 2019), BiGG (Dai et al., 2020), and BwR (Diamant et al., 2023) are auto-regressive models that sequentially generate graphs. GRAN uses attention-based GNNs to perform block-wise generation, focusing on dependencies between components within the graph. In contrast, BiGG addresses the challenges of efficiency by leveraging the sparsity of real-world graphs to avoid constructing dense representations. Unlike GRAN and BiGG, BwR simplifies the generation process further by restricting graph bandwidth. On the other hand, DiGress (Vignac et al., 2022) and HSpectra (Bergmeister et al., 2023) are built based on diffusion frameworks. DiGress is the first approach that uses a discrete diffusion model to iteratively modify graphs, while HSpectra focuses on multi-scale graph construction by progressively generating graphs through localized denoising diffusion.

**Molecule generation datasets.** We compare G2PT against four baselines: DiGress (Vignac et al., 2022), DisCo (Xu et al., 2024), Cometh (Siraudin et al., 2024), and DeFoG (Qin et al., 2024). Among them, DisCo and Cometh are both based on a continuous-time discrete diffusion framework, with Cometh additionally incorporating positional encoding for nodes and separate noising processes for nodes and edges. DeFoG adopts a discrete flow matching approach with a linear interpolation noising process.

**Graph pre-training methods.** We compare against several pre-training approaches for molecular property prediction, as summarized in Table 5. The goal ofGraph pre-training methods is to learn robust graph representations via exploiting the structural information. AttrMask (Hu et al., 2020a) uses attribute masking at both node and graph levels to capture local and global features simultaneously. ContextPred (Hu et al., 2020a) builds on this idea by predicting subgraph contexts, enabling the model to understand patterns beyond individual attributes. Similarly, InfoGraph (Sun et al., 2020) focuses on multi-scale graph representations by maximizing mutual information between graph-level embeddings and substructures. Moving to contrastive learning approaches, GraphCL (You et al., 2021) applies graph augmentations to generate positive and negative samples for representation learning. Building on this idea, GraphMVP (Liu et al., 2022a) incorporates both 2D molecular topology and 3D geometric views, aligning them within a contrastive framework to enhance feature representation. In contrast to these methods, GraphMAE (Hou et al., 2022b) adopts a generative approach, using a masked graph auto-encoder to reconstruct node features and capture structural information.

### B.6. Evaluation

**Metrics for molecule datasets.** As MOSES and GuacaMol are established benchmarking tools, they provide predefined metrics for evaluating and reporting results. These metrics are briefly outlined as follows: Validity assesses the percentage of molecules that satisfy basic valency constraints. Uniqueness evaluates the fraction of molecules represented by distinct SMILES strings, indicating non-isomorphism. Novelty quantifies the proportion of generated molecules absent from the training dataset. The filter score represents the percentage of molecules that satisfy the same filtering criteria applied during test set construction. The Frechet ChemNet Distance (FCD) (Preuer et al., 2018) quantifies the similarity between molecules in the training and test sets based on neural network-derived embeddings. SNN computes the similarity to the nearest neighbor using the Tanimoto distance. Scaffold similarity compares the distributions of Bemis-Murcko scaffolds, and KL divergence measures discrepancies in the distribution of various physicochemical descriptors.

For QM9 dataset, the validity metric reported in this study is calculated by constructing a molecule using RDKit and attempting to generate a valid SMILES string from it, as this approach is commonly employed in the literature. However, as explained by Jo et al. (2022b), this method has limitations, as it may classify certain charged molecules present in QM9 as invalid. To address this, they propose a more lenient definition of validity that accommodates partial charges, offering a slight advantage in their computations.

**Metrics for generic graph datasets.** We adopt the evaluation framework outlined by (Martinkus et al., 2022) and (Bergmeister et al., 2024), incorporating both dataset-agnostic and dataset-specific metrics. The dataset-agnostic metrics evaluate the alignment between the distributions of the generated graphs and the training data by analyzing general graph properties. Specifically, we characterize graphs based on their node degrees (Deg.), clustering coefficients (Clus.), orbit counts (Orbit), eigenvalues of the normalized graph Laplacian (Spec.), and statistics derived from a wavelet graph transform (Wavelet). To quantify the alignment, we compute the distances between the empirical distributions of these statistics for the generated and test graphs using Maximum Mean Discrepancy (MMD).

Subsequently, we evaluate the generated graphs using dataset-specific metrics under the V.U.N. framework, which measures the proportions of valid (V), unique (U), and novel (N) graphs. Validity is determined by dataset-specific criteria: graphs must be planar, tree-structured, or statistically consistent with a Stochastic Block Model (SBM) for the planar, tree, and SBM datasets, respectively. Uniqueness evaluates the proportion of non-isomorphic graphs among the generated samples, while novelty quantifies the proportion of generated graphs that are non-isomorphic to any graph in the training set.

### B.7. Computation Resources.

We ran all pre-training tasks and all goal-oriented generation fine-tuning tasks run on 8 NVIDIA A100-SXM4-80GB GPU with distributed training. For PPO training and graph property prediction tasks, we ran experiments using a A100 GPU.

## C. Extended Related works

### C.1. Auto-regressive Graph Generative Models

Even though graph is naturally an unordered set, auto-regressive models generate graphs sequentially, one node, edge, or substructure at a time. GraphRNN and DeepGMG (You et al., 2018b; Li et al., 2018) prefix a canonical ordering (e.g., breath-first search) for the nodes and edges and generates nodes and edges associated with them step by step. On the contrary,

---

**Algorithm 4** Depth-First search edge order generation

---
**Input:** Graph $G = (V, E)$, neighborhood function Nei.$(\cdot)$.
**Output:** Sequence of traversed edges $\sigma_E$.
Initialize $\sigma_E \leftarrow [\,]$, sample $v_0$ from $V$.

**DFS_helper** $(v)$:
**for** $v' \in$ Nei(v) **do**
  $e = (v, v')$.
  **if** $v'$ is not visited **then**
    Append $e$ to $\sigma_E$.
    Call **DFS_helper**$(v')$.
  **else**
    **if** $e \notin \sigma_E$ **then**
      Append $e$ to $\sigma_E$.
    **end if**
  **end if**
**end for**

Run **DFS_helper**$(v_0)$.

---

**Algorithm 5** Breadth-First Search edge order generation

---
**Input:** Graph $G = (V, E)$, neighborhood function Nei$(\cdot)$.
**Output:** Sequence of traversed edges $\sigma_E$.
Initialize $\sigma_E \leftarrow [\,]$, sample $v_0$ from $V$, initialize queue $\leftarrow [v_0]$.
**while** queue is not empty **do**
  $v \leftarrow$ queue.popfirst()
  **for** $v' \in$ Nei$(v)$ **do**
    $e = (v, v')$.
    **if** $v$ is not visited **then**
      append $e$ to $\sigma_E$, append $v'$ to queue.
    **else**
      **if** $e \notin \sigma_E$ **then**
        append $e$ to $\sigma_E$.
      **end if**
    **end if**
  **end for**
**end while**

---

Bacciu et al. (2020) propose to generate edges first then the connected nodes subsequently. These auto-regressive models are also broadly adapted into applications such as molecule generation. GCPN (You et al., 2018a), and REINVENT (Olivecrona et al., 2017) both leverage pre-trained auto-regressive models to fine-tune with a reward model to generate molecules with desired properties.

### C.2. Non-auto-regressive Graph Generative Models

In addition to auto-regressive models, non-auto-regressive graph generative models can be categorized into two branches: (1) one-shot generation and (2) iterative refinement. One-shot generation aims to generate a graph in a single step including methods such as generative adversarial networks (De Cao & Kipf, 2018), variational auto-encoders (Simonovsky & Komodakis, 2018; Liu et al., 2018) and normalizing flows (Madhawa et al., 2019; Zang & Wang, 2020). Nevertheless, one-shot graph generative models often suffer from the decoding strategies such that it requires an expressive decoder to map from latent vectors to graphs. On the other side, iterative refinement methods generate the entire graph in the first step and then iteratively refine the generated graph to be close to a realistic graph, including diffusion (Niu et al., 2020; Jo et al., 2022a; Vignac et al., 2022; Chen et al., 2022b; 2023; Jo et al., 2023; Haefeli et al., 2022; Wu et al., 2023; Siraudin et al.,

---

**Algorithm 6** Uniform edge order genration

---

**Input:** Graph $G = (V, E)$
**Output:** Sequence of edge ordering $\sigma_E$
Initialize $\sigma_E \leftarrow [\,]$
**while** $E$ is not empty **do**
    sample $e$ from $E$, append $e$ to $\sigma_E$
    Remove $e$ from $E$
**end while**

---

| Model | Edge Orderings | Validity↑ | Unique.↑ | Novelty↑ | Filters↑ | FCD↓ | SNN↑ | Scaf↑ |
|---|---|---|---|---|---|---|---|---|
| | Degree-based | 95.1 | **100** | 91.7 | 97.4 | 1.1 | 0.52 | 5.0 |
| G2PT$_{small}$ | DFS | 91.6 | **100** | 87.1 | 98.0 | 1.2 | **0.55** | 8.9 |
| | BFS | **96.2** | **100** | 86.8 | **98.3** | **1.0** | **0.55** | 10.6 |
| | Uniform | 62.9 | **100** | **99.4** | 52.0 | 7.0 | 0.38 | 9.5 |
| | Degree-based | 96.4 | **100** | 86.0 | 98.3 | **0.97** | **0.55** | 3.3 |
| G2PT$_{base}$ | DFS | 91.9 | **100** | 83.7 | 98.1 | 1.13 | **0.55** | 7.5 |
| | BFS | **96.9** | **100** | 84.6 | **98.7** | 0.98 | **0.55** | **11.1** |
| | Uniform | 80.9 | **100** | **97.0** | 83.9 | 2.14 | 0.46 | 10.3 |

*Table 9.* Sensitivity analysis on edge orderings.

2024; Xu et al., 2024; Wang et al., 2025) and flow matching models (Qin et al., 2024; Eijkelboom et al., 2024; Lipman et al., 2022; Liu et al., 2022b; Campbell et al., 2024; Gat et al., 2024). As discussed in **??**, they often require a prefixed number of refinement steps and they need to maintain an adjacency matrix over the trajectory which is computationally intensive.

### C.3. Pre-training Transformers for Graphs

Transformers are now dominating domains of natural language processing (NLP) and computer vision (CV) (Radford, 2018; Devlin, 2018; Dosovitskiy, 2020). Several works also attempt to applying transformers in the field of graph learning (Ying et al., 2021; Hu et al., 2020b; Dwivedi & Bresson, 2020; Rampášek et al., 2022; Chen et al., 2022a; Wu et al., 2021; Kreuzer et al., 2021; Min et al., 2022; Chen et al., 2024; Gao et al., 2024). Those approaches propose several methods to encode the graph structure information into sequences, specifically, the key research problem lies in how to add identifiers to nodes and tokenize the edges. For instance, Kim et al. (2022) uses positional embedding to help transformer to identify nodes, and type embeddings to distinguish node and edge tokens. A more recent work, Gao et al. (2024), which focuses on molecule representation learning, uses a vocabulary to store all atom types and all possible bond types (same bonds with different atoms as endpoints are considered as different type in their case). In contrast, by introducing node index into the vocabulary, G2PT easily implements the edge tokenizations and node identifications.

## D. Additional Results

### D.1. Sensitivity Analysis of Edge Orderings

We investigate how the employed edge orderings will affect the generative performance of G2PT. Specifically, we consider four orderings: the reverse of edge-removal process (Alg. 1), DFS ordering (Alg. 4), BFS ordering (Alg. 5), and uniform ordering (Alg. 6). We train G2PT$_{small}$ and G2PT$_{base}$ on MOSES dataset and evaluate the performance.

**Result.** Table 9 reports the performance of different edge orderings. BFS and degree-based edge-removal orderings both exhibit superior results, while DFS orderings show moderate performance. Particularly, uniform ordering shows poor performance in capturing the sequence distribution. This result highlights the importance of choosing the right ordering families for generating sequences.

### D.2. Additional Visualizations

We further visualize the generic graph in Figure 4, and molecular graph in Figure 5. The results show that both G2PT$_{small}$ and G2PT$_{base}$ have the ability to capture the topological rules of the training graphs.

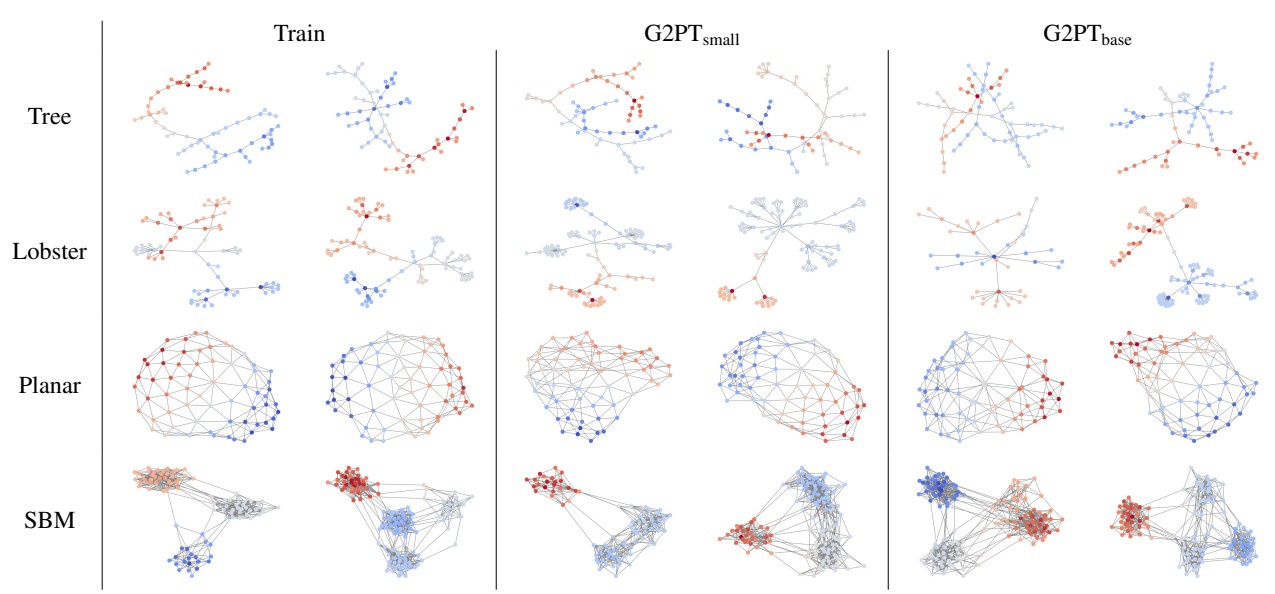

*Figure 4.* The visualization of generic graph datasets

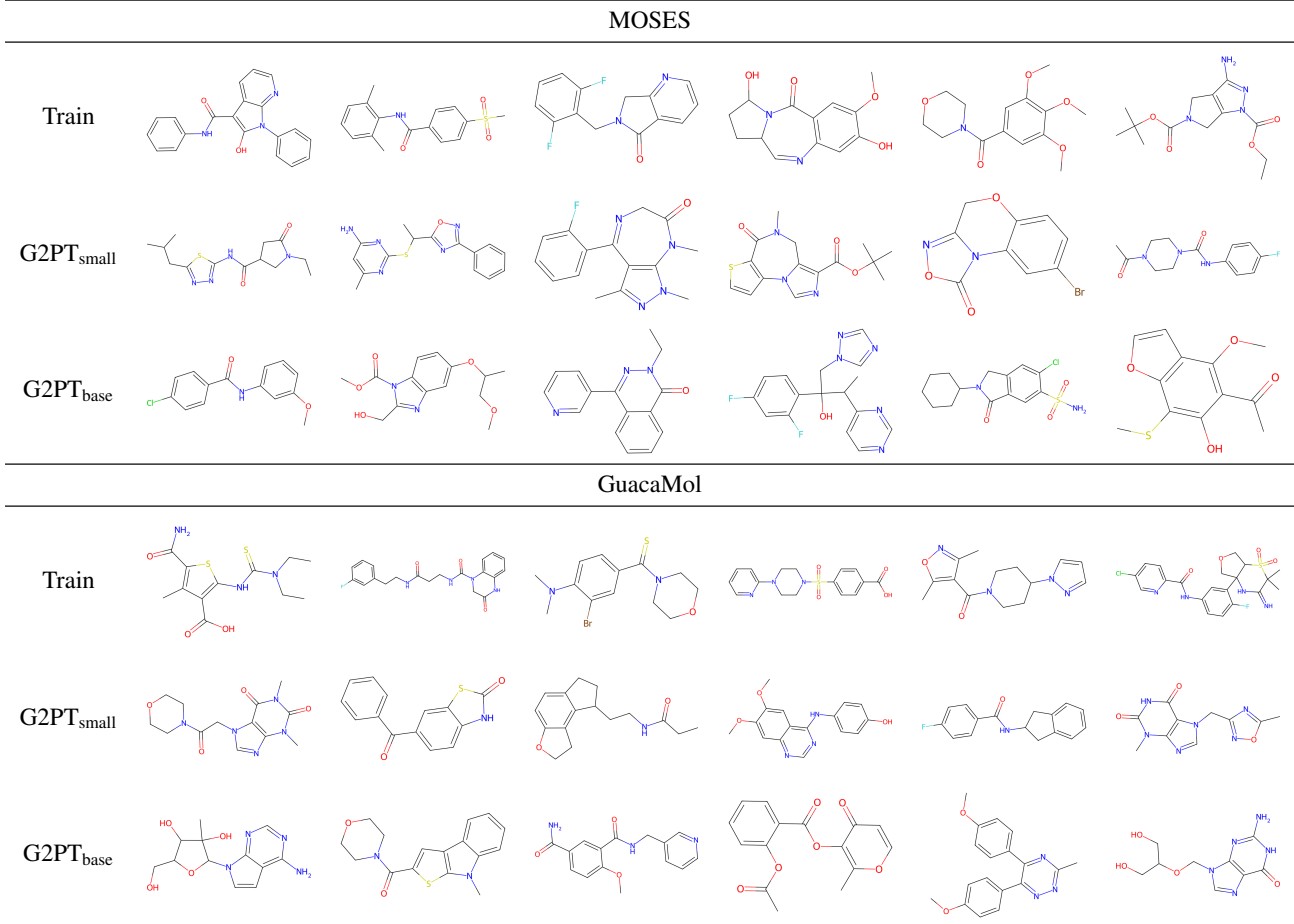

*Figure 5.* The visualization of molecular datasets

