# OpenReview forum: "Graph Generative Pre-trained Transformer"
_ICML.cc/2025/Conference — ICML 2025 poster_

### Official Review · Reviewer_jm1Y · 2025-03-13

**Overall Recommendation:** 4

**Summary:**

This paper introduces the graph generative pre-trained transformer (G2PT) for graph generation using auto-regressive transformers. The method introduces a sequence-based graph representation approach, which fits well to transformer architectures originally developed for NLP. The paper explores fine-tuning methods for downstream applications, e.g., goal-oriented graph generation and property prediction. Experiments demonstrate that G2PT achieves SOTA performance across several benchmarks, including generic graphs and molecular graphs.

**Claims And Evidence:**

Claims made in the submission look good to me.

**Essential References Not Discussed:**

I don't see essential references not discussed by the paper.

**Experimental Designs Or Analyses:**

1. I would suggest to add more detailed introduction/explanation for Table 2. From the title and text, it's not straightforward to know what tasks are performed there.

2. There are a lot of zeros in Table 2. Is it possible that the model overfits?

**Methods And Evaluation Criteria:**

Yes, they make sense to me.

**Other Comments Or Suggestions:**

NA

**Other Strengths And Weaknesses:**

The paper is easy to follow. The method proposed by the paper is straightforward yet effective, and complexity could be reduced by sequential representation compared to using adjacency matrix. The fine-tuning approaches (e.g., rejection sampling, reinforcement learning) are thoroughly explored and clearly demonstrated through experiments on goal-oriented molecule generation.

**Questions For Authors:**

Please see my comments above.

**Relation To Broader Scientific Literature:**

The graph generation task targeted by the paper is a big domain, including a few classic works such as GraphRNN, GRAN. Tokenizing graph is well motivated given the success of LLMs.

**Theoretical Claims:**

The major theoretical claim is that maximizing the sequence likelihood serves as maximizing a lower bound on the true graph likelihood, which is pretty clear to me.

---

> ### Author Rebuttal · Authors · 2025-03-30
>
> We thank the reviewer for the constructive suggestion! We address the concerns below.
>
> ---
>
> **Q1**. _I would suggest adding a more detailed introduction/explanation for Table 2. From the title and text, it's not straightforward to know what tasks are performed there._
>
> **A1**. Thanks for pointing this out! Since the experiment settings are standard and due to the page limit, we move most of the experiment details to the appendix. We will provide add a more detailed introduction for Table 2 in the next version.
>
> ---
>
> **Q2**. _​​There are a lot of zeros in Table 2. Is it possible that the model overfits?_
>
> **A2**. We report the V.U.N metric in our table. Specifically, this is computed by counting the percentage of generated graphs being valid, unique and novel. The novelty metric indicates whether the samples are different from the training graphs. Therefore, the model performs well not just by memorizing the training graph. Moreover, they are not exact zeros but we round it to 4 decimals (potentially a value of 1e-5  during evaluation). The values are so small may be due to the nature of  the graph statistics or simply because they are very easy to be captured by the model.
>
> ---
>
> We hope that we have addressed your concerns, thanks!

---

### Official Review · Reviewer_rB8b · 2025-03-13

**Overall Recommendation:** 2

**Summary:**

This paper proposes Graph Generative Pre-trained Transformer (G2PT) as a novel approach to molecular graph generation models. While conventional graph generation models are primarily adjacency matrix-based, this method treats node and edge lists as token sequences and employs an autoregressive Transformer (Transformer Decoder) for efficient learning. To extend G2PT as a general-purpose foundation model, it explores fine-tuning techniques for two downstream tasks: goal-oriented generation and graph property prediction. Computational experiments demonstrate that G2PT achieves state-of-the-art (SOTA) performance in molecular generation (QM9, MOSES, GuacaMol), general graph generation (Planar, Tree, Lobster, SBM), and downstream tasks (MoleculeNet).

**Claims And Evidence:**

The paper presents experimental results demonstrating that the proposed token sequence-based graph generation approach is more efficient than existing adjacency matrix-based methods. It also confirms that G2PT achieves performance comparable to or exceeding state-of-the-art (SOTA) models based on generation quality metrics such as MMD, Validity, Novelty, and FCD. Additionally, for goal-oriented generation, fine-tuning G2PT using rejection sampling (RFT) and PPO (reinforcement learning) effectively enhances target graph properties such as QED, SA, and GSK3β.

On the other hand, the way of representing graphs as token sequences is a very straightforward and cannot be said to build upon previous research. The extent to which "learning as a token sequence" and "learning through autoregressive next-token prediction" individually contribute remains unclear, and the evidence for each aspect appears somewhat insufficient. At the very least, there should be a discussion on how this method relates to existing token-based approaches—such as those highlighted in well-cited reviews like https://arxiv.org/abs/2302.04181—or, if specializing in molecular graphs, a comparison with simpler methods that feed traditional symbol-sequence representations of molecules like SMILES or SELFIES to Transformers.

**Essential References Not Discussed:**

There should be a discussion on how this method relates to existing token-based approaches—such as those highlighted in well-cited reviews like https://arxiv.org/abs/2302.04181—or, if specializing in molecular graphs, a comparison with simpler methods that just feed widely-established symbol-sequence representations of molecules like SMILES or SELFIES to Transformers.

Attending to Graph Transformers
https://arxiv.org/abs/2302.04181

SELFIES and the future of molecular string representations
https://doi.org/10.1016/j.patter.2022.100588

Transformer-based models for chemical SMILES representation: A comprehensive literature review
https://doi.org/10.1016/j.heliyon.2024.e39038

While this paper also evaluates generic tasks, its main focus is on molecular graph generation. Given that SMILES (and SELFIES) representations have a long history and are widely used for molecular representation, a natural question arises: why not simply input these representations into a Transformer? In fact, similar proposals have been repeatedly explored in the machine learning field, and discussions from the following paper and its peer reviews could provide useful insights. (**This doesn't mean the paper should cite the following papers; Just for information to make sure the main discussion points along this kind of research.**)

SmilesFormer: Language Model for Molecular Design
https://openreview.net/forum?id=VBQZkYu22G

SELFIES-TED : A Robust Transformer Model for Molecular Representation using SELFIES
https://openreview.net/forum?id=uPj9oBH80V

**Experimental Designs Or Analyses:**

The experimental datasets and tasks are well-balanced, incorporating molecular generation benchmarks such as MOSES and GuacaMol, as well as generic datasets. However, the QM9 dataset contains molecular data with 3D coordinates, which may not be the best fit for the proposed method. A more careful consideration of this choice may be necessary.

**Methods And Evaluation Criteria:**

While traditional graph generation models have primarily been adjacency matrix-based, this approach treats node and edge lists as token sequences and learns them efficiently using an autoregressive Transformer (Transformer Decoder). The paper notes that while early research on graph generation started with naive autoregressive methods, many recent high-performance models, such as those based on discrete diffusion, generate adjacency matrices. This makes it technically interesting to examine whether the autoregressive approach remains effective. Additionally, the study evaluates the method using two tasks: goal-oriented graph generation and property prediction. The evaluation also extends beyond molecular datasets to include generic datasets.

**Other Comments Or Suggestions:**

None

**Other Strengths And Weaknesses:**

None

**Questions For Authors:**

None

**Relation To Broader Scientific Literature:**

While traditional graph generation models have primarily been adjacency matrix-based, this approach treats node and edge lists as token sequences and learns them efficiently using an autoregressive Transformer (Transformer Decoder). The paper notes that while early research on graph generation started with naive autoregressive methods, many recent high-performance models, such as those based on discrete diffusion, generate adjacency matrices. This makes it technically interesting to examine whether the autoregressive approach remains effective.

**Theoretical Claims:**

N/A

---

> ### Author Rebuttal · Authors · 2025-03-30
>
> We thank the reviewer for the insightful comments. We believe the suggestion are very constructive in improving the quality of our draft, below we address the raised concern.
>
> ---
>
> **Q1**. _On the other hand, the way of representing graphs as token sequences is very straightforward and cannot be said to build upon previous research. The extent to which "learning as a token sequence" and "learning through autoregressive next-token prediction" individually contribute remains unclear, and the evidence for each aspect appears somewhat insufficient._
>
> **A1**. Yes, the way we represent graphs as tokens is both straightforward and novel as it is not built upon previous research. This is because previous token-based approaches have focused on **learning representations for graph/node/edge**, while ours focus on **generating graphs**.
>
> Regarding the contributions of each component, obtaining the token sequence is a necessary first step for applying next-token prediction. As a result, it is challenging to separate their individual impacts. Moreover, none of the earlier token-based methods are suitable for next-token prediction learning, which is why we developed this new tokenization approach. We will further elaborate on this in our response to your next question.
>
> ---
>
> **Q2**. _At the very least, there should be a discussion on how this method relates to existing token-based approaches - such as those highlighted in well-cited reviews like https://arxiv.org/abs/2302.04181._
>
> **A2**. Thank you for sharing the comprehensive survey covering previous graph-based transformer methods. We have thoroughly reviewed these approaches. Below we summarize what distinguish our method from them.
>
> - **Task**: G2PT is designed specifically for graph generation, whereas the token-based approaches discussed in the review primarily target graph representation learning. This fundamental difference necessitates distinct tokenization strategies for our task.
>
> - **Tokenization**: Correspondingly, G2PT’s tokenization is **invertible** between graph and token sequence. That said, we can obtain the graph by de-tokenizing the graph. While the previous token-based approaches only address how to represent graphs into tokens (node / node+edge / subgraph), translating token sequence back to graph remains unclear.
>
> GPT-like models have demonstrated tremendous success across various domains. However, attempts on utilizing them in graph generation remain inadequate. We believe the barrier lies in how to efficiently tokenize and de-tokenize graphs, which our work directly addresses.
>
> ---
>
> **Q3**. _or, if specializing in molecular graphs, a comparison with simpler methods that feed traditional symbol-sequence representations of molecules like SMILES or SELFIES to Transformers._
>
> **A3**. G2PT is more generic in representing graphs, making it  more flexible thus more suitable for various molecule-related applications compared to traditional symbol-sequence. Such applications include **constrained generation**, **molecule in-painting**, **retrosynthesis**, etc. (https://arxiv.org/pdf/2502.09571, https://arxiv.org/abs/2308.16212)
>
> In contrast, since symbol-sequences like SMILES/SELFIES are canonical for graphs, modifying any node or edge on the graphs will lead to a transformative change on the sequence's representation. Methods that operate on SMILES/SELFIES need to employ a seq-to-seq approach to generate a new molecule from scratch given the condition one.
>
> G2PT's principle of tokenizing a graph is more general and can be further extended beyond the one we define in the paper. By introducing addition/deletion/replace actions, alteration can be easily performed on the original molecule via defining a sequence of action. Such action trajectory well aligns with the process of lead optimization. And we believe such agentic paradigm would be more universal and scalable.
>
> ---
> **Q4**. _The experimental datasets and tasks are well-balanced, incorporating molecular generation benchmarks such as MOSES and GuacaMol, as well as generic datasets. However, the QM9 dataset contains molecular data with 3D coordinates, which may not be the best fit for the proposed method. A more careful consideration of this choice may be necessary._
>
> **A4**. We agree that QM9 contains 3D information. The reason we chose QM9 is to follow prior works (DiGress, DeFoG, Cometh) to provide a more comprehensive comparison.
>
> ---
>
> We hope that we have addressed your concern. Let us know if you have any following-up questions.

---

> > ### Comment · Reviewer_rB8b · 2025-04-08
> >
> > Thank you for your comment. The additional information has partially addressed my concerns, and I now have a better understanding of the paper's contributions. At the same time, because the proposed approach is quite simple, I also feel that the paper should have at least included these detailed discussions and/or experimental comparisons with existing tokenization methods and traditional linear molecular representations like SMILES and SELFIES. If the proposed tokenization method is specifically designed for generation (rather than molecular representation learning itself), providing evidence to support that would make it both interesting and valuable. I look forward to seeing future improvements.

---

> > > ### Author Response · Authors · 2025-04-09
> > >
> > > Thanks for your response! For sure that we are happy to further address your concern.
> > >
> > > ---
> > >
> > > **Q1.** _I also feel that the paper should have at least included these detailed discussions and/or experimental comparisons with existing tokenization methods and traditional linear molecular representations like SMILES and SELFIES_
> > >
> > > **A1.** We consider two baselines: **GEEL** and **LigGPT**. Here GEEL proposes another graph tokenization approach and LigGPT is based on SMILES sequences. We compare G2PT against these two baselines on molecular datasets. Note that due to the limited time, we copy the performance of LigGPT from its paper. For GEEL, we report the reproduced results, which we've done earlier per Reviewer TGsg's request.
> > >
> > > - Paper links:
> > >
> > > GEEL: https://arxiv.org/pdf/2312.02230
> > >
> > > LigGPT: https://chemrxiv.org/engage/chemrxiv/article-details/60c7588e469df48597f456ae
> > >
> > > - Results on MOSES
> > >
> > > ||Validity	|Unique	|Novelty	|Filters	|FCD	|SNN	|Scaf	|
> > > |-|-|-|-|-|-|-|-|
> > > |GEEL	|92.1	|100	|81.1	|97.5	|1.28	|0.52	|**3.6**	|
> > > |LigGPT	|90.0	|99.9|**94.1**	|-	|-	|-	|-	|
> > > |G2PT	|**96.4**	|**100**	|86.0	|**98.3**	|**0.97**	|**0.55**	|3.3	|
> > >
> > >
> > > - Results on GuacaMol
> > >
> > > ||Validity|Unique|Novelty|KL Div.|FCD|
> > > |-|-|-|-|-|-|
> > > |GEEL|88.2|98.2|89.1|93.1|71.5|
> > > |LigGPT|**98.6**|99.8|**100**|-	|-	|
> > > |G2PT|94.6|**100**|99.5|**96.0**|**93.4**|
> > >
> > > G2PT achieves superior result compared to the two baselines. **Note that LigGPT performs temperature tuning (1.6 for MOSES and 0.9 for GuacaMol) to maximize performance while ours use the default temperature (1.0) for sampling.**
> > >
> > > We hope the comparison has addressed your concerns. We will include those results along with the discussion in previous response in our updated draft.
> > >
> > > ---
> > >
> > > **Q2.** _If the proposed tokenization method is specifically designed for generation (rather than molecular representation learning itself), providing evidence to support that would make it both interesting and valuable_
> > >
> > > **A2.** Thank you for the insightful question. Indeed we develop this tokenization approach for generation task while previous approaches mostly focus on representation learning. However, whether the G2PT's tokenization could be used for representation learning remains unexplored. While we explore on a downstream graph prediction task in the submission, we believe there maybe a better paradigm (training objective) for G2PT tokenization.
> > >
> > > One of the direction we are currently exploring is to apply encoder-based transformer for mask modeling. This is implicitly performing node reconstruction and edge prediction task when trying to recover token from mask. We are still working on introducing informative learning tasks in graph-level. And since such work deviates the problem we targeted to address in the submission, we will leave such work in the future.
> > >
> > > ---
> > >
> > > We hope our follow-up responses addressed your remaining concerns. Thank you again for taking the time to review our submission and rebuttal.

---

### Official Review · Reviewer_TGsg · 2025-03-14

**Overall Recommendation:** 4

**Summary:**

Authors introduce a new way to represent graph as sequence of tokens, that contains both node definitions and edge definitions. They use this representation and standard transformer architecture trained on next-token prediction task to generate new graphs. The method is competetive to SOTA diffusion and non-autoregressive graph generative methods, which is not the case for older autoregressive approaches. Authors further expand their methods by showing its utility for downstream molecule property prediction tasks as well as RL finetuning for goal oriented generation.

## update after rebuttal

The authors adressed my main concerns raised, and overall I think this is a good paper. Thus I recommend acceptance

**Claims And Evidence:**

I find all claims to be well validated experimentally using established benchmarks.

**Essential References Not Discussed:**

Authors failed to discuss and compare to probably the most comparable work in the literature to theirs: https://arxiv.org/pdf/2312.02230 which also proposes a new way to represent graphs for autoregresive generation.

**Experimental Designs Or Analyses:**

All experiments follow standard practices and use standard datasets.

**Methods And Evaluation Criteria:**

Authors use a wide range of well established graph generation benchmarks that are good for testing graph generation approaches.

**Other Comments Or Suggestions:**

A small note is that in lines 327-328 authors quote DiGress as source for valid, unique and novel samples (V.U.N.) metrics, while they were originally introduced in SPECTRE (https://arxiv.org/abs/2204.01613).

**Other Strengths And Weaknesses:**

No other weaknesses. The method is quite straight forward, the validation is quite extensive and the results look good and are in line with other SOTA approaches, which is good enough for a method resurecting a much older way of doing things (autoregressive generation).

**Questions For Authors:**

I'd like to see a comparison to https://arxiv.org/pdf/2312.02230.

Can the authors also expand on tie-breaking in Algorithm 1? How is the problem solved when multiple nodes have the same degree? Are ties broken randomly and ordering is different in each epoch for the same sample or are they broken in a fixed maner based on original graph IDs.

**Relation To Broader Scientific Literature:**

The paper revisits autoregressive graph generation and shows that with modern neural netrowks and smarter data representation the autoregressive methods can be competetive to the current SOTA diffusion-based graph generative methods. They do skip one very relevant related work as per below.

**Theoretical Claims:**

The theoretical claims are quite standard and rely on well known methods and thus are sound.

---

> ### Author Rebuttal · Authors · 2025-03-30
>
> We thank the reviewer for the constructive review, below we address the raised question/comment.
>
> ---
> **Q1**. _Discussion with GEEL (https://arxiv.org/pdf/2312.02230) and comparison_
>
> **A1**. We first provide a discussion with GEEL then provide the experimental result of G2PT and GEEL on 6 graph datasets.
>
> - Discussion:
>
> G2PT and GEEL both transform graphs into edge lists. Typically, an edge list is viewed as a sequence of node pairs, such as $[(s_1, t_1), (s_2, t_2), …, (s_m, t_m)]$. The efficiency of this representation hinges on the construction of the vocabulary. A basic method treats each pair as a distinct “word,” leading to a vocabulary size on the order of $O(N^2)$ for a graph with N nodes, which becomes unscalable and directly depends on the graph size.
>
> To address this, GEEL reduces the vocabulary size by drawing inspiration from the concept of graph bandwidth B. This idea is based on the observation that, after an appropriate node permutation, only the entries near the diagonal of the adjacency matrix are nonzero, thereby shrinking the vocabulary from $O(N^2)$ to $O(B^2)$.
>
> In contrast to GEEL, G2PT represents each node pair using two tokens rather than a single token. This multi-token approach provides better flexibility and avoids any assumptions about the graph’s structure, reducing the vocabulary size significantly to $O(N)$.
>
> It is also important to note that for any graph, a trivial lower bound for the graph bandwidth is given by $bw(G) ≥ \Delta/2$, where $\Delta# is the maximum degree of the graph. Thus, while GEEL’s vocabulary size may vary depending on the graph’s pattern, G2PT’s approach remains pattern-agnostic.
>
> - Result Comparison: We compare G2PT with GEEL on generic graphs (Planar, Tree, Lobster, SBM) and molecular graphs (MOSES, GuacaMol).
>
> _Planar_
>
> |      | Deg | Clus   | Orbit  | Spec   | Wavelet | V.U.N. |
> |------|--------|--------|--------|--------|---------|--------|
> | GEEL | **1e-3** | 1e-2 | 1e-3 | - | - | <27.5 |
> | G2PT | 1.8e-3 | **4.7e-3** | **0.00** | 8.1e-3 | 5.1e-3 | **100** |
>
>
> _Tree_
>
> |      | Deg | Clus   | Orbit  | Spec   | Wavelet | V.U.N. |
> |------|--------|--------|--------|--------|---------|--------|
> | GEEL | **1.5e-3** | **0.00** | 2e-4 | 1.5e-2 | **4.6e-3** | 90 |
> | G2PT | 4.2e-3 | **0.00** | **1e-4** | **7.3e-3** | 5.7e-3 | **99** |
>
>
> _Lobster_
>
> |      | Deg | Clus   | Orbit  | Spec   | Wavelet | V.U.N. |
> |------|--------|--------|--------|--------|---------|--------|
> | GEEL | 2e-3 | **0.00**  | 1e-3 | - | - | <72.7 |
> | G2PT | **1e-3** | **0.00** | **0.00** | 4e-3 | 1e-2 | **100** |
>
>
> _SBM_
>
> |      | Deg | Clus   | Orbit  | Spec   | Wavelet | V.U.N. |
> |------|--------|--------|--------|--------|---------|--------|
> | GEEL | 2.5e-2 | **3e-3** |2.6e-2 | - | - | <42.5 |
> | G2PT | **4.2e-3** | 5.3e-3 | **3e-4** | 6.1e-3 | 6.9e-3  | **100** |
>
>
> _MOSES_
>
> |      | Validity | Unique   | Novelty  | Filters   | FCD | SNN | Scaf |
> |------|--------|--------|--------|--------|---------|--------|--------|
> | GEEL | 92.1 | **100** | 81.1 | 97.5 | 1.28 | 0.52 | **3.6** |
> | G2PT | **96.4** | **100** | **86.0** | **98.3** | **0.97** | **0.55** | 3.3 |
>
> _GuacaMol_
>
> |      | Validity | Unique   | Novelty  | KL Div.   | FCD |
> |------|--------|--------|--------|--------|---------|
> | GEEL | 88.2 | 98.2 | 89.1 | 93.1 | 71.5 |
> | G2PT | **94.6** | **100** | **99.5** | **96.0** | **93.4** |
>
> We will include the discussion as well as the experiment result in our next version.
>
> ---
>
> **Q2**. _V.U.N reference correction_
>
> **A2**. Thanks for correcting this! We will update the reference in the next version.
>
> ---
>
> **Q3**. _Can the authors also expand on tie-breaking in Algorithm 1? How is the problem solved when multiple nodes have the same degree? Are ties broken randomly and ordering is different in each epoch for the same sample or are they broken in a fixed manner based on original graph IDs._
>
> **A3**. Excellent question! Yes we break the tie randomly and will obtain a different order every epoch. We observe that using more than 10 orders for every graph leads to better generalization and performance (similar to data augmentation). We kindly refer to the reviewer in Figure 3 for how using more orders affects the overall validity of molecular graphs.
>
> ---
>
> Let us know whether we address your concerns, thanks!

---

### Decision · Program_Chairs · 2025-05-01

**Decision:**

Accept (poster)

**Comment:**

This paper focuses on graph generation.  The paper proposes a straight-forward yet effective method for graph generation.  The approaches centers on a new way to represent a graph as sequence of tokens, that contains both node definitions and edge definitions.  Generation is achieved via an auto-regressive framework on this representation.

The paper had divergent reviews after the rebuttal phase.  The reviewers appreciated the simplicity and effectiveness of the framework.  The results on standard benchmarks (Planar, Tree, Lobster, and SBM) were solid.  Concerns were about the results on molecular datasets and comparisons to standard string-based graph representations.

On balance, the contribution is sufficient for ICML, though with concerns over the overall magnitude of contribution.